# SRPK1 acetylation modulates alternative splicing to regulate cisplatin resistance in breast cancer cells

Cheng Wang[1], Zhihong Zhou[2], Charannya Sozheesvari Subhramanyam[1], Qiong Cao[1], Zealyn Shi Lin Heng [1], Wen Liu [3], Xiangdong Fu [4] & Qidong Hu [1✉]

Cisplatin and other platinum-based compounds are frequently used to treat breast cancer, but their utility is severely compromised by drug resistance. Many genes dictating drug responsiveness are subject to pre-mRNA alternative splicing which is regulated by key kinases such as the serine-arginine protein kinase 1 (SRPK1). However, its contribution to drug resistance remains controversial. In this study, we have identified that Tip60-mediated acetylation of SRPK1 is closely associated with chemotherapy sensitivity. In breast cancer cells, cisplatin induced SRPK1 acetylation but in the corresponding resistant cells, it reduced acetylation yet increased phosphorylation and kinase activity of SRPK1, favouring the splicing of some anti-apoptotic variants. Significantly, the cisplatin-resistant cells could be re-sensitized by enhancing SRPK1 acetylation or inhibiting its kinase activity. Hence, our study reveals a key role of SRPK1 in the development of cisplatin resistance in breast cancer cells and suggests a potential therapeutic avenue for overcoming chemotherapy resistance.

[1] Department of Anatomy, Yong Loo Lin School of Medicine, National University of Singapore, 4 Medical Drive, Singapore, Singapore 117594. [2] Department of Physiology, Yong Loo Lin School of Medicine, National University of Singapore, 2 Medical Drive, Singapore, Singapore 117593. [3] School of Pharmaceutical Sciences, Fujian Provincial Key Laboratory of Innovative Drug Target Research, Xiamen University, Xiang'an South Road, Xiamen, Fujian 361102, China. [4] Department of Cellular and Molecular Medicine, University of California, San Diego, 9500 Gilman Drive, La Jolla, CA 92093-0651, USA. ✉email: anthq@nus.edu.sg

**B**reast cancer is highly heterogeneous and the optimal therapy depends on the molecular and histological signatures of individual cases[1]. For triple-negative breast cancer (TNBC) that lacks oestrogen receptor (ER), progesterone receptor (PR) and epidermal growth factor receptor 2 (HER2) amplification, chemotherapy is used either alone or in adjuvant or neoadjuvant settings[2]. Cisplatin is a prototype of platinum-based chemodrugs that can form platinum-DNA adducts to initiate apoptosis[2,3]. However, a major hurdle that prevents its long-term use is the drug resistance acquired by tumour cells[4].

Cisplatin resistance has been attributed to altered pharmacokinetic factors, enhanced DNA damage repair, inactivation of pro-apoptotic and activation of pro-survival signalling[3–7]. Remarkably, many genes involved in these processes are modulated by pre-mRNA alternative splicing (AS) which is capable of generating multiple isoforms from a single locus[8–10]. The family of serine/arginine-rich splicing factors (SRSFs) is an important group of regulatory proteins for AS[11]. Their functions are mediated through the phosphorylation-dephosphorylation cycle by several kinase families. One consists of the serine-arginine protein kinases 1–3 (SRPK1–3). SRPK1 is the most extensively studied and known to target RS dipeptides in SRSFs to facilitate their nuclear translocation[12,13]. In the nucleus, SRSFs are further phosphorylated, preferentially at SP dipeptides, by another kinase family, CDC2-like kinases 1–4 (CLK1–4)[13,14]. On the other hand, SRPK1 also translocates into the nucleus to strip CLK1 from the fully phosphorylated SRSFs, a critical event to regulate pre-mRNA AS[13].

Although the aberrant expression of SRPK1 has been recorded in many cancers, including lung, prostate, male germ cells, retinoblastoma, pancreas, colon and breast, the kinase is controversially related to both chemotherapy sensitivity and resistance[15–21]. More surprisingly, SRPK1 expression is associated with either cisplatin sensitivity or resistance in the ovarian cancer[22,23]. Hence, it is conceivable that SRPK1 may be differentially modulated in distinct tumours, and the protein level is not the only determinant for its role in cisplatin responsiveness. Indeed, SRPK1 itself is subject to autophosphorylation which enhances its kinase activity and nuclear translocation[24]. Multiple signalling cascades, including osmotic stress, EGF-EGFR-Akt and CK2, promote this post-translational modification (PTM) of SRPK1[24–26].

Apart from phosphorylation, an acetylome analysis revealed that lysine acetylation could be another critical PTM for proteins involved in RNA splicing, such as SRPK1[27]. Another study found that Lysine Acetyltransferase 5 (KAT5/Tip60) could acetylate SRSF2, thus affecting its protein turnover and downstream AS events[28]. Interestingly, Tip60 expression was correlated with cisplatin resistance in prostate, epidermoid and lung cancer cells[29]. However, in breast cancer cells, whether SRPK1 can indeed be acetylated, whether the acetylation is mediated by Tip60 and whether this novel PTM of SRPK1 could lead to cisplatin resistance remain unknown.

In this study, we aimed to elucidate how the PTMs of SRPK1 contribute to the acquisition of cisplatin resistance. We noted that cisplatin increased SRPK1 acetylation in a Tip60-dependent manner in breast cancer cells. Surprisingly, in contrast to prostate and epidermoid cancers[29], the acquisition of cisplatin resistance in breast cancer cells is associated with a decrease in Tip60 expression and SRPK1 acetylation. This led to enhanced phosphorylation of SRPK1 and SRSFs, and induced AS switch of some critical genes involved in apoptosis towards the anti-apoptotic variants. Importantly, the cisplatin-resistant cells could be re-sensitized by enhancing the Tip60-dependent acetylation of SRPK1 or suppressing its kinase activity with a specific inhibitor, SRPIN340. These findings suggest a feasible strategy to improve the effectiveness of platinum-based chemotherapy to treat breast cancers.

## Results

**SRPK1 expression in breast cancer cells**. The cisplatin sensitivity of breast cancer cells, MCF7 and MDA-MB-231 (231), and the corresponding chemoresistant lines, MCF7R and 231R, was evaluated by MTS viability assays. As indicated by the IC50 values, MCF7R and 231R were about three to five times more resistant to cisplatin than the parental lines (Fig. 1a, b). As a confirmation, cisplatin dramatically increased phospho-H2AX (pH2AX)[30] in MCF7 and 231 cells, but not in MCF7R or 231R (Supplementary Fig. 1).

To explore whether drug resistance is linked to AS, we first checked the expression of SRPK1. Immunoblotting showed that while there was a decrease in SRPK1 protein level in cisplatin-treated MCF7 cells, the kinase was upregulated by the drug in 231 cells (Fig. 1c; Supplementary Fig. 2). Interestingly, SRPK1 expression was not affected by the treatment in either MCF7R or 231R cells, but was generally higher in MCF7R than MCF7, and lower in 231R than 231 (Fig. 1c; Supplementary Fig. 2). These observations were confirmed in the cells treated with cisplatin in a time course for up to 5 days (Supplementary Fig. 3). Surprisingly, although SRPK1 knockdown moderately decreased the IC50 in MCF7R, its overexpression showed no effect in 231R (Fig. 1d, e), implying that the expression level of SRPK1 may not be the only major determinant of cisplatin responsiveness.

**SRPK1 is acetylated in a Tip60-dependent manner**. To resolve the conundrum, we proceeded to examine the PTMs of SRPK1. The kinase was overexpressed with wild-type Tip60 or a HAT-deficient mutant (Q377E/G380E), a dominant-negative form of Tip60[31]. The mass spectrometry analysis predicted five lysine residues in SRPK1 that could be acetylated in Tip60-overexpressing cells, 215, 258, 265, 301 and 318 K (Supplementary Fig. 4; Supplementary Data 1). Indeed, these sites are also predicted by PAIL (Prediction of Acetylation on Internal Lysines) and dbPTM. Curiously, we did not detect acetylation at 585 or 588 K as reported in acute myeloid leukaemia cells[27], which could be due to different cellular contexts. Nevertheless, to explore the biological relevance of SRPK1 acetylation, the five newly identified lysine residues together with 585 and 588 K were mutated to arginine and the resultant mutant was designated as Mut7. Myc-tagged SRPK1 or Mut7 was next introduced into Tip60-overexpressing cells, and the acetylation deficiency in Mut7 was confirmed by immunoprecipitation with a c-Myc antibody (Fig. 2a). Reciprocal experiment showed that SRPK1 was abundantly acetylated in Tip60-expressing cells, but not in HAT-deficient Tip60 mutant-transfected cells (Fig. 2b). We further observed that the HDAC inhibitor, Trichostatin A (TSA), dramatically enhanced SRPK1 acetylation but Tip60 knockdown abolished this (Fig. 2c–f). Interestingly, cisplatin also increased SRPK1 acetylation, which was again abolished by Tip60 depletion (Fig. 2g). These observations indicate that Tip60 is involved in SRPK1 acetylation.

To explore whether cisplatin resistance is related to SRPK1 acetylation, we checked Tip60 expression and noted a cisplatin-induced increase in both MCF7 and 231, but not in MCF7R or 231R (Fig. 2h; Supplementary Fig. 5a). Additionally, immunofluorescence revealed that although Tip60 is well recognized as a histone acetyltransferase, a considerable amount of the protein was localized in the cytoplasm (Supplementary Fig. 5b). Another two platinum-based compounds, carboplatin and oxaliplatin, also upregulated Tip60 in 231, but to a much lesser extent in 231R cells which exhibited cross-resistance (Supplementary Fig. 6).

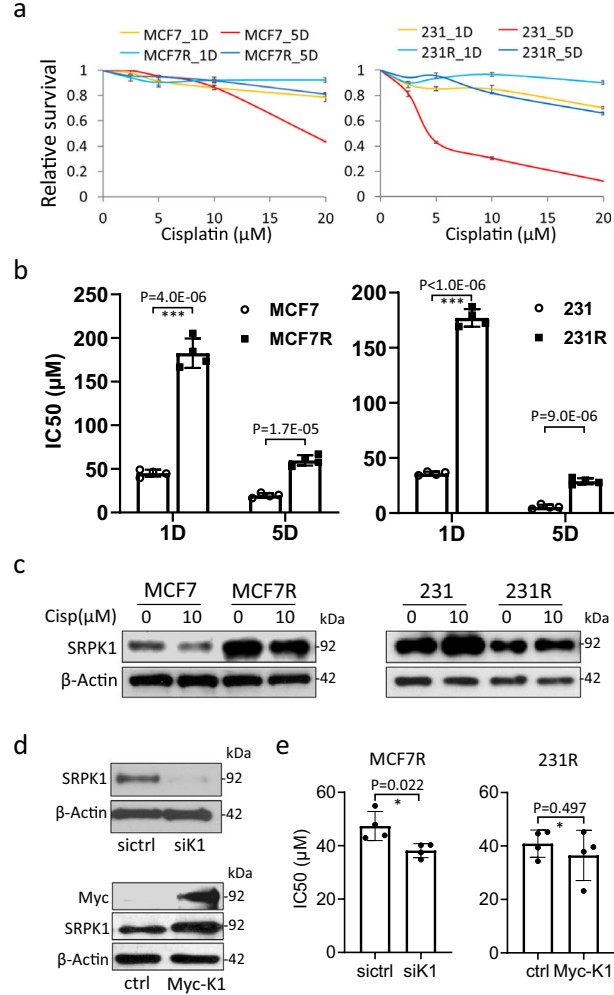

**Fig. 1 Profiling of SRPK1 expression in the parental and cisplatin-resistant breast cancer cells. a** MCF7, MCF7R, 231 and 231R cells were treated with DMF or different concentrations of cisplatin for 1 day (1D) or 5 days (5D). The cell survival was then assessed by the MTS viability assay. The reading of cisplatin-treated cells was normalized against that of DMF-treated cells. Data points: mean ± SD; $n = 3$. **b** The IC50 of cisplatin for the indicated cell lines was derived from the above MTS viability assay using the Hill equation. Bars: mean ± SD; $n = 4$; ***$p < 0.001$ by Student's $t$-test. D: days of cisplatin treatment. **c** The indicated cells were treated with DMF or 10 μM cisplatin for 5 days. Immunoblotting was then performed with the SRPK1 antibody. **d** The knockdown efficiency of the siRNA SMARTpool targeting SRPK1 (siK1) and overexpression of Myc-tagged SRPK1 (Myc-K1) were verified by immunoblotting. **e** MCF7R cells were transfected with siK1 and 231R with Myc-K1, and subjected to MTS viability assays. The IC50 of cisplatin was derived using the Hill equation. Bars: mean ± SD; $n = 4$; *$p < 0.05$ by Student's $t$-test. Cisp: cisplatin. sictrl: negative control siRNA. ctrl: control pCMV-myc vector. The Western blots in (**c**) and (**d**) are representative of three and four experiments with similar results, respectively.

Consistently, immunoprecipitation revealed that SRPK1 acetylation was increased by cisplatin in both MCF7 and 231, but decreased in MCF7R and 231R (Fig. 2i), suggesting that a low level of SRPK1 acetylation may confer chemoresistance.

**Acetylation interferes with SRPK1 and SRSF phosphorylation.** Given that SRPK1 is subject to autophosphorylation[25,32], we first examined the relationship between the two PTMs. Immunoprecipitation showed that Tip60 increased SRPK1 acetylation while

decreased its phosphorylation in MCF7, and that the HAT-deficient Tip60 could restore the phosphorylation status (Fig. 3a), supporting an inverse correlation between these two PTMs of SRPK1.

As established previously[25], we next confirmed that over-expressed SRPK1 could phosphorylate SRSFs by using an antibody (mAb104) that recognizes multiple phosphorylated SRSFs[24]. The phosphorylation was even enhanced by over-expressing Mut7 instead, especially for SRSF1/2 and SRSF5 (Fig. 3b). This could be explained by the acetylation deficiency that led to elevated phosphorylation status and kinase activity of Mut7. Indeed, as cisplatin reduced SRPK1 acetylation in MCF7R (Fig. 2i), we noted a dose-dependent increase in SRPK1 phosphorylation (Fig. 3c) and more prominent SRSF phosphorylation in cisplatin-treated MCF7R than MCF7 (Fig. 3d). To test whether SRPK1 acetylation could compromise SRSF activation, we overexpressed Tip60 in MCF7R and noticed a drastic decrease in SRSF phosphorylation (Fig. 3e).

An elevated level of SRPK1 phosphorylation was also observed in 231R in which SRPK1 was less acetylated (Figs. 2i, 3f). Indeed, immunoprecipitation confirmed that the acetylation-deficient Mut7 was more phosphorylated than the wild-type counterpart (Fig. 3g). Since SRPK1 autophosphorylation could be enhanced by AKT[24], we next checked the activation status of AKT. Intriguingly, AKT-Ser473 was more phosphorylated in 231R than in 231, which was not accompanied by an elevated phosphoryla-tion of mTOR-Ser2448, a major target of AKT activation[33] (Supplementary Fig. 7), supporting the previous report that SRPK1 could be another downstream effector of AKT independent of mTOR[24].

As in MCF7R (Fig. 3d), SRSFs were generally more phosphorylated in cisplatin-treated 231R than 231 (Fig. 3h), which could be reduced by overexpressing the wild-type, but not the HAT-deficient, Tip60, especially for SRSF4 and SRSF6 (Fig. 3i). Interestingly, when Tip60 was co-expressed with Mut7, SRSFs were still substantially phosphorylated (Fig. 3j). These findings strongly support the notion that acetylation could interfere with SRPK1 phosphorylation and its subsequent activation of SRSFs.

**Nuclear localization of SRPK1 in cisplatin-resistant cells.** Phosphorylation can promote SRPK1 nuclear translocation[25,34,35]. In agreement with a higher level of SRPK1 phosphorylation in cisplatin-resistant cells (Fig. 3f), a more pro-minent SRPK1 staining was found in the nuclei of MCF7R and 231R than MCF7 and 231, respectively (Fig. 4a; Supplementary Fig. 8a). Immunoblotting confirmed that SRPK1 was amply present in the nuclear fraction of 231R (Supplementary Fig. 8b). Moreover, co-immunoprecipitation revealed a stronger interac-tion between SRPK1 and heat shock protein 90 (HSP90) in 231R than in 231 (Fig. 4b, c), which is consistent with the notion that HSP90 facilitates SRPK1 nuclear translocation[24].

To examine whether acetylation could affect SRPK1 localiza-tion, a GFP-tagged SRPK1 or Mut7 was transiently expressed. While GFP-SRPK1 was mostly cytoplasmic, GFP-Mut7 was readily detected in the nucleus of HeLa (Fig. 4d; Supplementary Fig. 9a) and MCF7 cells (Supplementary Fig. 9b). Fractionation of SRPK1- or Mut7-transfected cells confirmed that Mut7 was more prone to be localized in the nucleus (Fig. 4e). As a result, potent phosphorylation of splicing factors, especially SRSF1/2 and SRSF6, was also noted in the nuclear fraction of Mut7-transfected cells (Fig. 4e). Moreover, HAT-deficient Tip60 increased SRPK1 nuclear localization, but Mut7 remained substantially nucleus-localized even with wild-type Tip60 over-expression (Supplementary Fig. 9c). These observations indicate

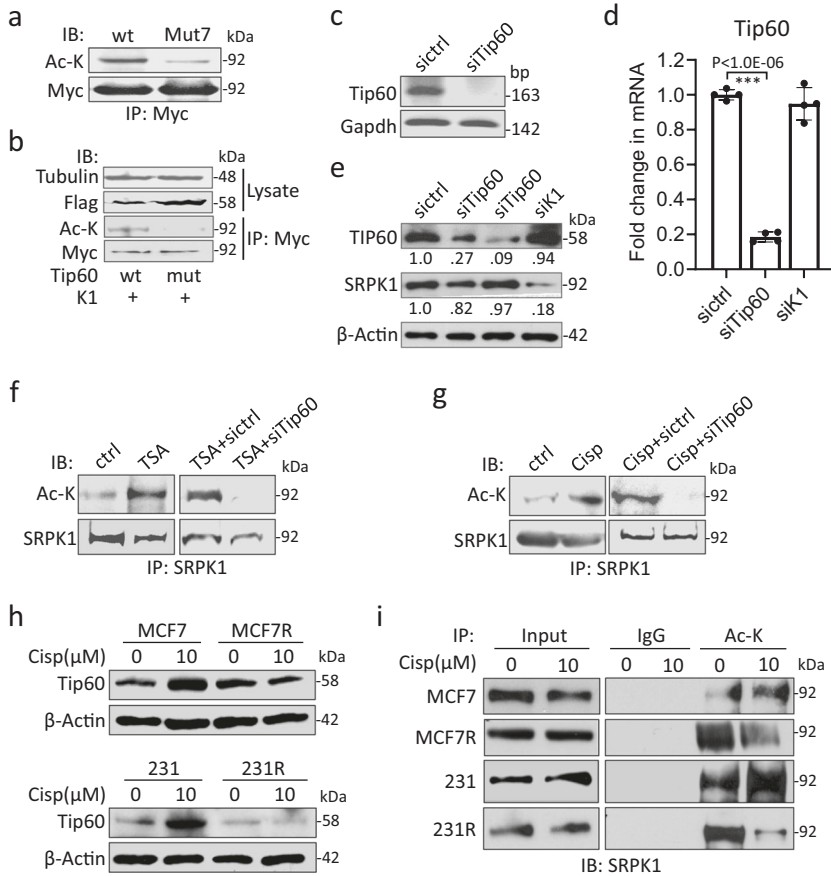

**Fig. 2 SRPK1 was acetylated in a Tip60-dependent manner. a** Myc-tagged SRPK1 (wt) or acetylation-deficient SRPK1 (Mut7) was overexpressed in 293T cells. Immunoprecipitation (IP) was then performed using the c-Myc antibody, and the precipitates were immunoblotted (IB) with the acetyl-lysine antibody (Ac-K). **b** Myc-tagged SRPK1 (K1) was co-expressed with either Flag-tagged wild-type (wt) or HAT-deficient (mut) Tip60. The immunoprecipitation and immunoblotting were then performed with indicated antibodies. **c–e** The knockdown efficiency of the siRNA pool targeting Tip60 (siTip60) was verified by RT-PCR in 293T cells (**c**), by qPCR (**d**) and immunoblotting (**e**) in 231 cells. Bars in (**d**): mean ± SD; $n = 4$; ***$p < 0.001$ by Student's $t$-test. The decimals below the strips in (**e**) denote th**e** relative abundance of Tip60 and SRPK1. **f**, **g** 293T cells were then transfected with sictrl or siTip60, and treated with TSA (**f**) or cisplatin (**g**). The acetylation of SRPK1 was examined by immunoprecipitation with the SRPK1 antibody and immunoblotting with Ac-K. **h** The indicated cells were treated with 10 μM cisplatin for 5 days and the protein level of Tip60 was checked by immunoblotting. **i** The acetylation of SRPK1 in the indicated cells was assessed by immunoprecipitation with Ac-K followed by immunoblotting for SRPK1. For IP experiments, 5 μM TSA was added to the medium 6 hours before harvest. For RT-PCR gel images in (**c**) and Western blots in (**a**, **b**, **e–g**, **i**), they are representative of three experiments with similar results. The Western blots in (**h**) are representative of four experiments with similar results.

that SRPK1 acetylation may prevent its nuclear translocation, thus affecting its function in activating SRSFs.

**Acetylation reduces the stability of SRPK1**. As previously reported, Tip60 can target SRSF2 and induce its proteasome-dependent degradation[28]. Interestingly, with the protein translation inhibited by cycloheximide (CHX), we observed a time-dependent decrease of SRPK1 in Tip60-expressing 293T cells, but not in those expressing the HAT-deficient Tip60 mutant (Fig. 5a). When co-expressed with Tip60, SRPK1 exhibited a time-dependent decrease while Mut7 remained stable (Fig. 5b). Consistently, the cytoplasmic staining of SRPK1 was dramatically decreased by Tip60 overexpression, but Mut7 was still strongly immunolabelled in the nucleus (Fig. 5c; Supplementary Fig. 10), suggesting that acetylation deficiency may preserve the protein stability of SRPK1.

Indeed, as cisplatin increased Tip60 expression and SRPK1 acetylation in MCF7 and 231 (Fig. 2h, i), there was a time-dependent decrease of endogenous SRPK1 in both parental cells, which was not observed in either MCF7R or 231R (Fig. 5d). These observations support the notion that acetylation could reduce the

protein stability and consequently diminish the amount of SRPK1 available for phosphorylation and nuclear translocation.

**SRPK1 acetylation affects AS**. To investigate whether pre-mRNA AS is involved in regulating cisplatin sensitivity, we first checked the splicing of BARD1 (Supplementary Fig. 11a), a co-factor of BRCA1 involved in DNA damage repair[36,37]. The full-length (FL) and isoform α play a tumour suppressor role[36], and the isoform γ is suggested to stabilize FL[38]. RT-PCR showed that their abundance was increased by cisplatin in 231, but not in 231R. On the other hand, BARD1φ and BARD1δ were associated with poor survival in breast cancer patients[39,40], and they were elevated by cisplatin in 231R (Supplementary Fig. 11b). Interestingly, SRPK1 and Tip60 co-expression dramatically increased FL and BARD1α in 231R, which was abolished by replacing SRPK1 with Mut7, or Tip60 with the HAT-deficient mutant (Supplementary Fig. 11c, d).

We further studied the splicing of two members of the BCL2 family, BCL2L1 (Bcl-x) and MCL-1 (myeloid cell leukaemia-1)[41]. For both genes (Supplementary Fig. 12a, b), the long isoforms (Bcl-xL and MCL-1L) possess the anti-apoptotic activity while the

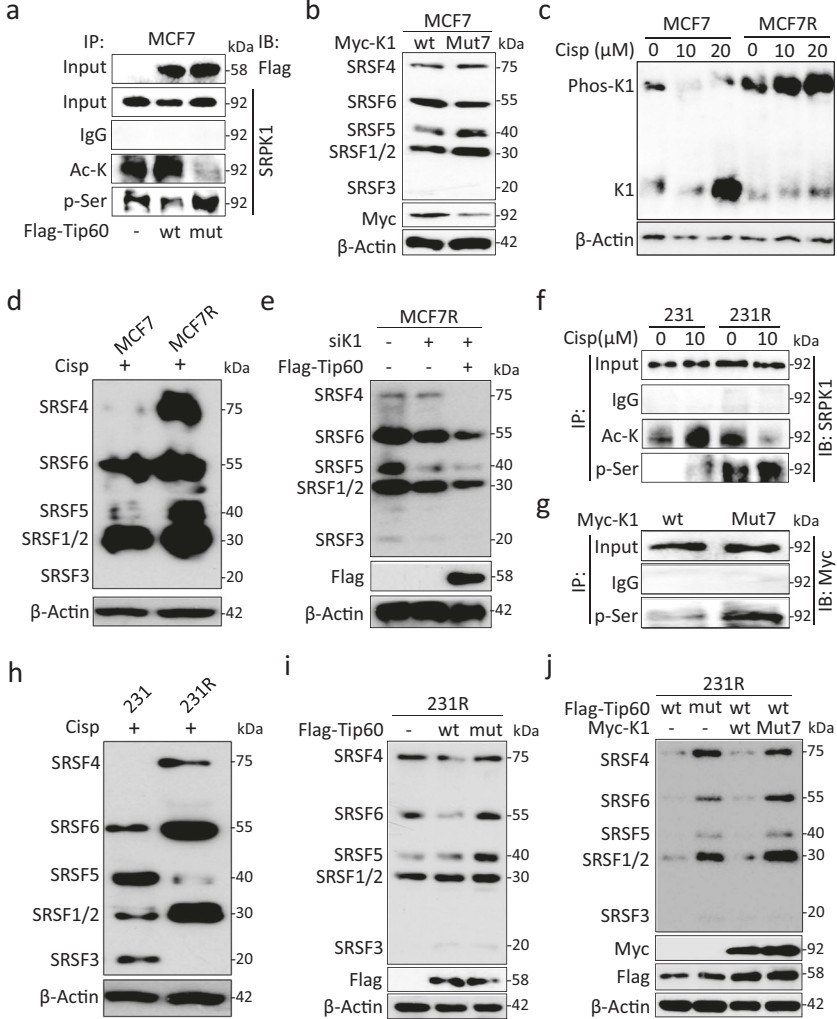

**Fig. 3 Acetylation affected phosphorylation of SRPK1 and SRSFs in cisplatin-resistant cells. a** MCF7 cells were transfected with Flag-tagged Tip60 (wt) or HAT-deficient mutant (mut). The PTMs of SRPK1 were assessed by immunoprecipitation with Ac-K and p-Ser antibodies. **b** MCF7 cells were transfected with Myc-tagged SRPK1 (wt) or Mut7. The phosphorylation of SRSFs was evaluated by using the mAb104 antibody. **c** MCF7 and MCF7R cells were treated with the indicated concentrations of cisplatin for 5 days and lysed for immunoblotting for SRPK1 (K1) using the Phos-tag gel. **d** In cisplatin-treated MCF7 and MCF7R cells, SRSF phosphorylation was detected by mAb104. **e** MCF7R cells were transfected with SRPK1 siRNA (siK1) with or without concurrent overexpression of Flag-tagged Tip60. The effect on SRSF phosphorylation was examined by mAb104. **f** 231 and 231R cells were treated with DMF or cisplatin. Then the relative levels of SRPK1 acetylation and phosphorylation were determined by using the Ac-K and p-Ser antibodies in immunoprecipitation. **g** 231 cells were transfected with Myc-tagged SRPK1 (wt) or Mut7. The immunoprecipitation was performed with the phospho-Ser beads, followed by immunoblotting with the c-Myc antibody. **h** In cisplatin-treated 231 and 231R cells, the phosphorylation of SRSFs was checked by mAb104. **i** Flag-tagged Tip60 (wt) or HAT-deficient mutant (mut) was introduced into 231R cells. Immunoblotting was then performed with mAb104. **j** Flag-tagged Tip60 constructs were co-expressed with Myc-tagged SRPK1 constructs as indicated. The SRSF phosphorylation level was evaluated with mAb104. For Phos-tag gel in (**c**) and Western blots in the other panels, they are representative of three experiments with similar results.

short isoforms (Bcl-xS and MCL-1S) function as pro-apoptotic factors[42–44]. By RT-PCR, we found that the relative abundance of Bcl-xS and MCL-1S was increased by cisplatin in 231, but decreased in 231R in a dose-dependent manner (Fig. 6a).

Then we asked whether SRPK1 acetylation could affect the AS profile of *BCL2L1* and *MCL-1*. RT-PCR showed that while SRPK1 or Tip60 overexpression alone marginally upregulated Bcl-xS and MCL-1S in cisplatin-treated 231R (Fig. 6b, lanes 2 and 4; Supplementary Fig. 12c), their concurrent overexpression dramatically increased the relative abundance of these pro-apoptotic variants, as well as the tumour suppressive BARD1-FL and BARD1α, which was reversed when SRPK1 acetylation was prevented (Fig. 6b, lanes 6 and 7; Supplementary Fig. 12c, lanes 5–7), suggesting that SRPK1 acetylation may favour pro-apoptotic splicing.

To confirm this notion, an mCherry-fused MCL-1 splicing reporter was utilized. Premature termination codons (PTCs) were inserted into the exon 2 which is skipped in MCL-1S[45]. When the reporter was introduced into cisplatin-treated SRPK1-overexpressing 231R cells, only about 20% cells displayed detectable mCherry signals, whereas ~60% cells co-expressing SRPK1 and Tip60 showed strong mCherry signals, indicating the production of pro-apoptotic MCL-1S (Fig. 6c). However, when Tip60 was co-expressed with Mut7, mCherry was barely detectable (Fig. 6c), suggesting that the acetylation deficiency in Mut7 resulted in less MCL-1S produced. Consistently, in 231 cells where SRPK1 acetylation was increased by cisplatin (Figs. 2i, 3f), when Tip60 or SRPK1 was knocked down (Supplementary Fig. 13a), the cisplatin-induced increase in pro-apoptotic MCL-1S was reversed (Supplementary Fig. 13b, lanes 3 and 4).

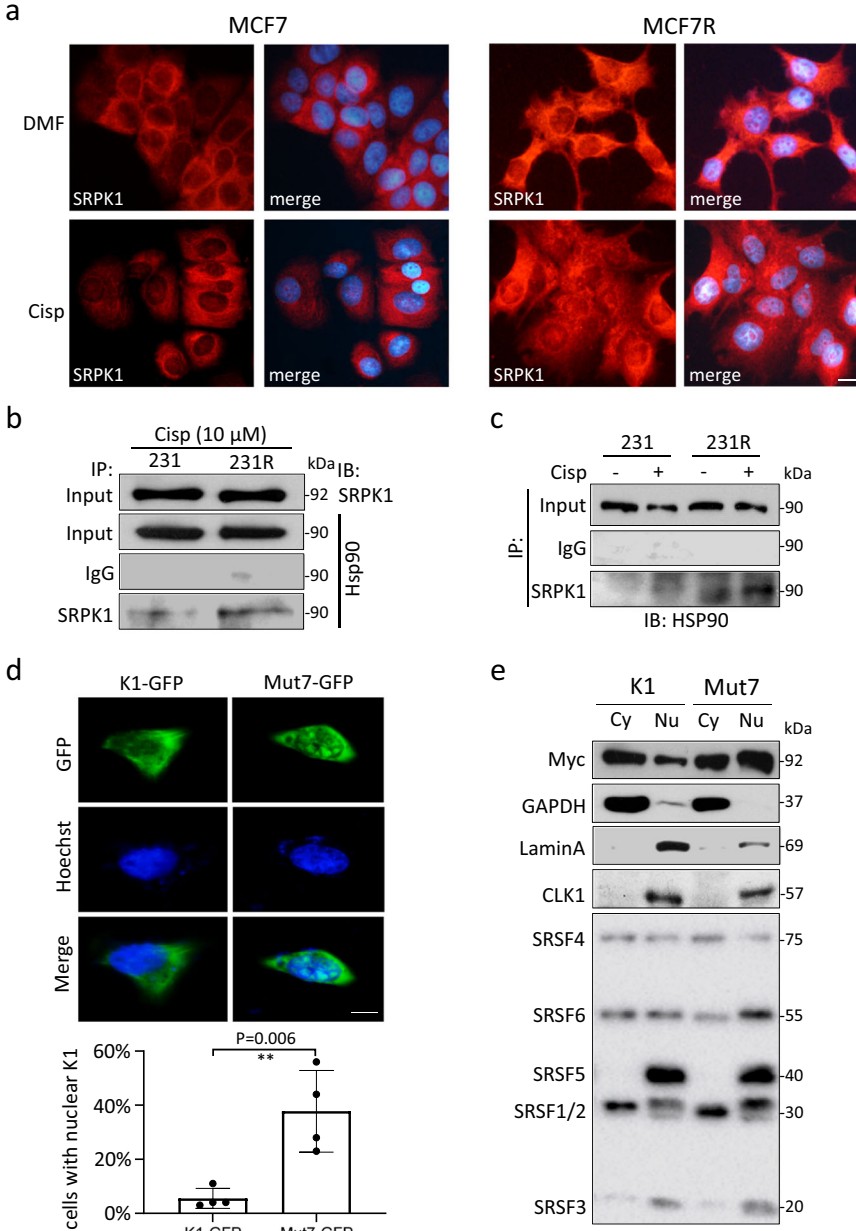

**Fig. 4 Acetylation affected the subcellular localization of SRPK1. a** MCF7 and MCF7R cells were treated with cisplatin for 5 days and immunostained for SRPK1. The nuclei were counterstained with Hoechst 33342. Scale bar: 20 μm. **b**, **c** 231 and 231R cells were treated with cisplatin and the potential interaction between SRPK1 and HSP90 was characterized by co-immunoprecipitation assays using the indicated antibodies. **d** HeLa cells were transfected with GFP-tagged SRPK1 or Mut7. The subcellular localization of GFP signals was examined by the fluorescence microscopy. Scale bar: 5 μm. Bars: mean ± SD; $n = 4$; **$p < 0.01$ by Student's $t$-test. **e** HeLa cells overexpressing Myc-tagged SRPK1 or Mut7 were fractionated. The relative abundance of Myc-SRPK1 and phosphorylated SRSFs in the cytoplasm (Cy) and nucleus (Nu) was determined by immunoblotting. GAPDH was used as a cytoplasmic marker. LaminA and CLK1 were markers for the nuclear fraction. Cisp: cisplatin. The immunofluorescence images in (**a**) are representative of four experiments with similar results. The Western blots in (**b**, **c**, **e**) are representative of three experiments with similar results.

Similarly, in cisplatin-treated MCF7R, the dominant variant expressed was MCL-1L, and a drastic switch towards the production of pro-apoptotic MCL-1S and Bcl-xS was observed when Tip60 was overexpressed and SRPK1 knocked down (Supplementary Fig. 13c, d, lane 5).

VEGF-A is a canonical SRPK1/SRSF1-regulated splicing target[46]. VEGF-A$_{165}$ is a pro-angiogenic splice variant whereas the short VEGF-A$_{165}$b variant with 66 bp excluded in exon 8 antagonizes angiogenesis[47] (Supplementary Fig. 14a). BAXα is another pro-apoptotic member of the BCL2 family and the splice variant β is more aggressive in triggering apoptosis, whereas

variants like δ and ζ lack the BH3 domain for homodimerization or heterodimerization with other BCL2 proteins[48] (Supplementary Fig. 14b).

Markedly, we did not detect the presence of VEGF-A$_{165}$b in either 231/231R or MCF7/MCF7R cells (Supplementary Fig. 14c, d). On the other hand, the prototypical BAXα was less produced in both MCF7R and 231R as compared with their respective parental cells (Supplementary Fig. 14c, e). Furthermore, cisplatin induced the splicing of the constitutively active BAXβ in 231 but not 231R cells (Supplementary Fig. 14c). These observations suggest that BAX expression could be synergistically regulated at

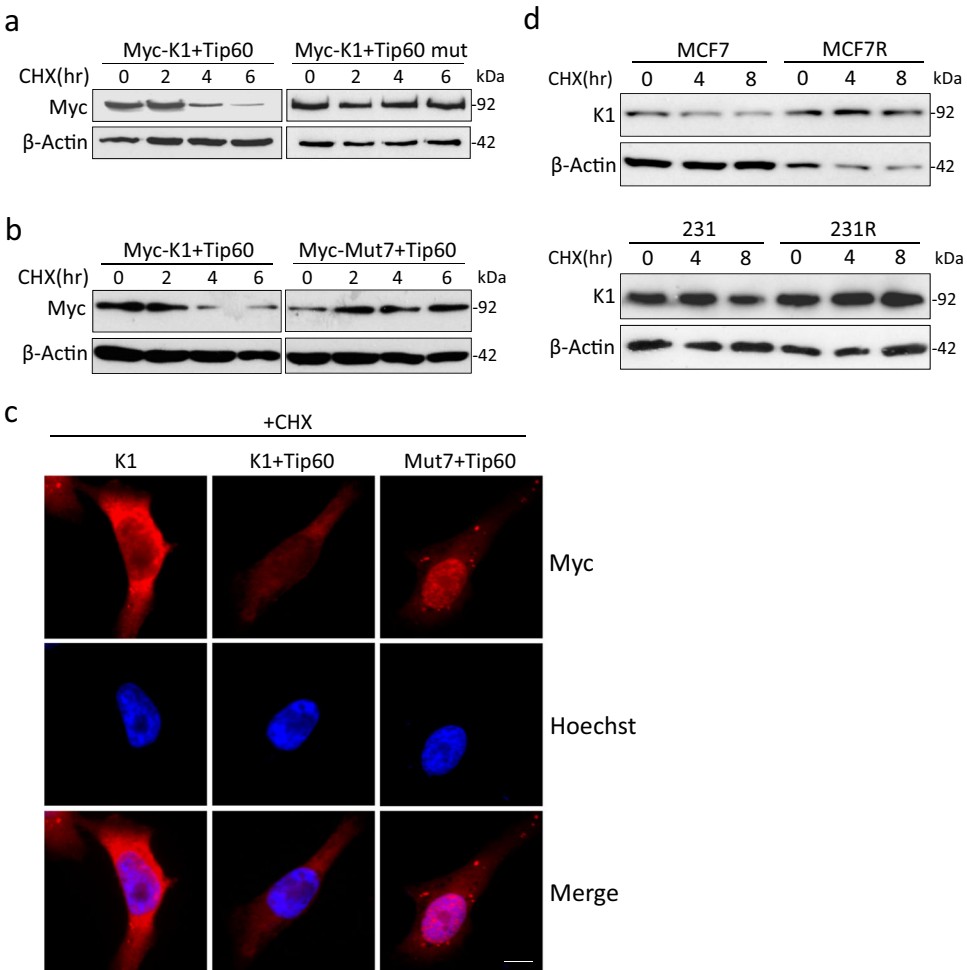

**Fig. 5 The protein stability of SRPK1 was affected by acetylation.** 293T cells were transfected with Myc-tagged SRPK1 (Myc-K1) and Tip60 or HAT-deficient mutant (mut) (**a**), or transfected with Tip60 and Myc-tagged SRPK1 or Mut7 (**b**). The cells were then treated with 100 µM CHX for indicated time. The level of Myc-tagged SRPK1 protein was checked by immunoblotting. **c** 293T cells were transfected with Myc-tagged SRPK1 (K1) or Mut7 and Tip60 as indicated. The cells were then immunostained with the c-Myc antibody. The nuclei were counterstained with Hoechst 33342. Scale bar: 5 µm. **d** The indicated cells were treated with cisplatin for 24 hours, followed by CHX for the indicated time. The protein level of endogenous SRPK1 was evaluated by immunoblotting. Western blots in (**a**, **b**, **d**) and immunofluorescence in (**c**) are representative of three experiments with similar results.

both the transcriptional level and splicing level. Notably, the BH3-deficient BAXδ was readily expressed in 231, MCF7 and MCF7R cells, but not in 231R cells. As the exact function of BAXδ remains to be characterized, future study is warranted to examine whether this specific variant is related to chemoresistance in a subset of breast cancers, such as TNBC as represented by 231 cells. Taken together, we propose that SRPK1 acetylation could affect the cisplatin effectiveness via modifying the splicing of some key regulators of apoptosis in breast cancer cells.

**SRPK1 acetylation is prevalent in TNBCs.** As chemotherapy remains a first-line treatment for TNBCs, we further utilized HCC70, BT549 and MDA-MB-468 (468) to validate if SRPK1 acetylation regulates drug sensitivity in these TNBC lines. Cisplatin was found to induce the acetylation of SRPK1 in BT549 and 468 cells, but not in HCC70 (Supplementary Fig. 15a). Hence, we proceeded to establish the cisplatin-resistant lines from parental BT549 and 468 cells, and 1.3 to 2-fold resistance to cisplatin was recorded in thus derived cells, which were designated as BT549R and 468R, respectively (Supplementary Fig. 15b). Immunoblotting showed that cisplatin did not consistently increase or decrease SRPK1 expression in either BT549/BT549R pair or 468/468R pair

(Supplementary Fig. 15c, d). However, Tip60 was upregulated by the cisplatin treatment in both parental lines, but remained uninduced in BT549R or moderately decreased in 468R (Supplementary Fig. 15c, d). As 468R showed twofold resistance to cisplatin as compared with the parental 468 (Supplementary Fig. 15b) and exhibited opposite response to the chemodrug in terms of Tip60 expression (Supplementary Fig. 15d), the 468/468R pair was chosen for further analysis. Like in the case of 231/231R (Fig. 3f), the acetylation of SRPK1 was induced by cisplatin only in 468, but not in 468R (Supplementary Fig. 15e). Moreover, cisplatin switched the splicing towards the pro-apoptotic Bcl-xS and MCL-1S specifically in 468 cells, which also expressed a higher level of the pro-apoptotic BAXα than 468R cells (Supplementary Fig. 15f). These observations indicate that the regulatory effect of SRPK1 acetylation on splicing might be applicable to other TNBC lines.

**SRPK1 acetylation re-sensitizes cisplatin-resistant cells.** Next we examined whether the drug sensitivity can be altered by manipulating SRPK1 acetylation. In MCF7 and 231 cells, Tip60 knockdown significantly elevated the IC50 of cisplatin, indicating fewer apoptotic cells (Supplementary Fig. 16a). Indeed, with Kaplan-Meier Plotter[49], a low expression of Tip60 was correlated

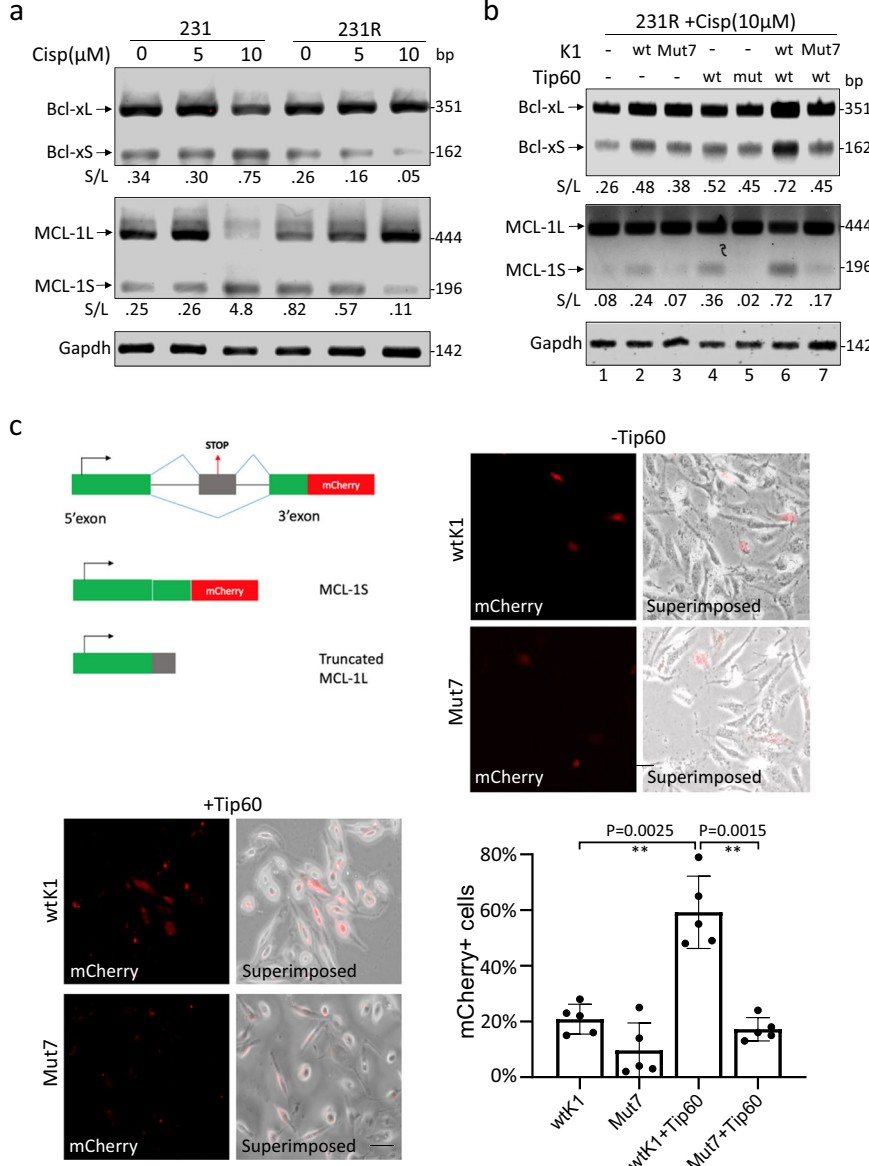

**Fig. 6 Alternative splicing of BCL-2 family genes could be affected by SRPK1 acetylation. a** 231 and 231R cells were treated with indicated concentrations of cisplatin. The alternative splicing of *BCL2L1* and *MCL-1* was examined by RT-PCR. **b** In cisplatin-treated 231R cells, the acetylation of SRPK1 was manipulated by the indicated single transfection and co-transfection. The levels of alternatively spliced variants of *BCL2L1* and *MCL-1* were checked by RT-PCR. The decimals below the gel strips in (**a**, **b**) denote the relative abundance of short (S) versus long (L) variants. **c** 231R cells were co-transfected with the mCherry-fused MCL-1 splicing-sensitive reporter (MCL1-PTC mCherry), Tip60 and SRPK1 or Mut7 as indicated. The mCherry signals were recorded by the fluorescence microscopy and superimposed onto the phase-contrast images. Scale bar: 20 μm. Bars: mean ± SD; $n = 5$; **$p < 0.01$ by Student's $t$-test. The RT-PCR gel images in (**a**, **b**) are representative of three experiments with similar results.

with poorer survival of breast cancer patients (Supplementary Fig. 16b). Reciprocally, in MCF7R, when Tip60 overexpression was coupled with SRPK1 knockdown, the IC50 was significantly decreased, which was abolished if Tip60 was replaced with the HAT-deficient mutant (Fig. 7a, bars 5 and 6). On the other hand, as SRPK1 was lower in 231R than in 231 (Fig. 1c; Supplementary Figs. 2, 3), the IC50 of cisplatin in 231R was reduced by co-expression of Tip60 with SRPK1, which was again abolished when SRPK1 was replaced with Mut7 (Fig. 7b, bars 6 and 7). Consistently, intensive pH2AX labelling was noted when SRPK1 and Tip60 were co-expressed in 231R, implying massive DNA damage (Supplementary Fig. 16c). With double immunolabelling, stronger pH2AX signals were observed in 231R cells

overexpressing SRPK1 than in those expressing Mut7 (Supplementary Fig. 16d), suggesting that SRPK1 acetylation could ultimately affect the repair of the cisplatin-induced DNA lesion.

To confirm this notion, we assessed cell apoptosis by examining PARP1 cleavage (cPARP1)[50]. As expected, it was induced by cisplatin in MCF7 and 231 cells, but not in MCF7R or 231R (Fig. 7c, d, lanes 1–4). Interestingly, Tip60 overexpression combined with SRPK1 knockdown elevated PARP1 cleavage in cisplatin-treated MCF7R (Fig. 7c, lane 8). In 231R cells, co-expression of Tip60 with SRPK1, but not Mut7, also induced PARP1 cleavage (Fig. 7d, lanes 8 and 9). These observations confirm that SRPK1 acetylation is involved in regulating the cellular response to cisplatin.

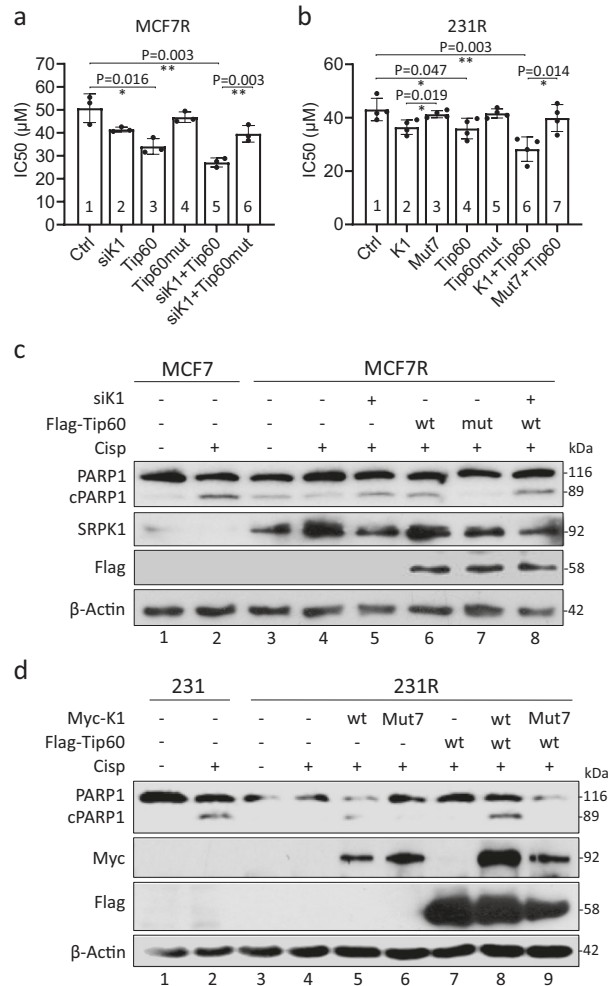

**Fig. 7 The acetylation of SRPK1 reduced IC50 and increased apoptosis in cisplatin-resistant cells.** The acetylation level of SRPK1 in MCF7R (**a**) and 231R (**b**) was manipulated by the indicated transfection. The cells were then subjected to MTS viability assays to obtain the IC50 of cisplatin. Bars: mean ± SD; $n = 3$ (**a**) and $n = 4$ (**b**); *$p < 0.05$, **$p < 0.01$ as determined by Student's $t$-test. MCF7/MCF7R (**c**) and 231/231R cells (**d**) were treated with DMF or 10 μM cisplatin for 5 days and transfected with siRNAs or overexpressing plasmids as indicated. The cell lysates were then analysed by immunoblotting for PARP1 and the cleaved PARP1 (cPARP1). The Western blots in (**c**, **d**) are representative of three experiments with similar results.

**Inhibition of SRPK1 activity restores cisplatin sensitivity.** Since acetylation was inversely correlated with SRPK1 phosphorylation and the subsequent activation of SRSFs (Fig. 3), we next explored whether inhibiting the kinase activity of SRPK1 directly with SRPIN340[51] could affect the splicing of apoptosis regulators and thus the cisplatin sensitivity. First, the efficacy of SRPIN340 was confirmed by a dose-dependent decrease in SRSF phosphorylation in cisplatin-treated 231R (Fig. 8a). MTS assays then showed that SRPIN340 reduced the IC50 of cisplatin in 231R (~40%) and in MCF7R (~25%) at 10 μM (Fig. 8b). Hence, when the cells were co-treated with SRPIN340 and cisplatin, PARP1 cleavage (Fig. 8c) and the relative abundance of pro-apoptotic Bcl-xS and MCL-1S (Fig. 8d; Supplementary Fig. 17) were increased. Consistently, in 231R cells expressing the MCL-1 splicing reporter, while the production of pro-apoptotic mCherry-fused MCL-1S was not induced by cisplatin alone, it was potently switched on by the addition of SRPIN340 (Fig. 8e). Hence, by targeting the PTMs of SRPK1, we

may alter the pre-mRNA AS of key regulators of apoptosis and modulate the cisplatin sensitivity of breast cancer cells.

## Discussion

For breast cancer, especially TNBC, the platinum-based compounds remain an important treatment option[2]. However, the long-term effectiveness is often undermined by drug resistance acquired by tumour cells[4].

Recent studies have indicated that pre-mRNA AS could be a unifying mechanism to modulate drug pharmacokinetics, DNA damage repair and pro-apoptotic/anti-apoptotic signalling[8,9]. In particular, aberrant expression of SRPK1, a key kinase for RNA splicing, has been observed in a wide spectrum of malignancies including breast, prostate, colon, lung, ovarian, male germ cell and retinoblastoma[15–23,52,53]. However, its expression seems to be related to both chemotherapy sensitivity and resistance, raising an intriguing possibility that other properties of SRPK1 may be critical in regulating cellular response.

Remarkably, a genome-wide acetylome study has unravelled a significant overrepresentation of the proteins involved in RNA splicing, including SRSFs and SRPKs[27]. Indeed, our study reveals that SRPK1 acetylation has a significant impact on the splicing of key regulators of apoptosis in breast cancer cells. Moreover, the acetylation was dependent on Tip60 which was universally upregulated by cisplatin, carboplatin and oxaliplatin in sensitive cells, but not in cisplatin-resistant cells (Supplementary Fig. 6). As a result, the resistant breast cancer cells could be re-sensitized by enhancing Tip60-mediated SRPK1 acetylation (Fig. 7). However, a previous study shows that in lung carcinoma cells, cisplatin decreases both the transcript and protein levels of Tip60, resulting in nuclear translocation of SRPK1 and SRPK2[28]. Indeed, a high Tip60 expression level has been related to cisplatin resistance in prostate, epidermoid and lung cancer cells[29]. This discrepancy may be due to different cellular contexts, including crosstalk between splicing factors and upstream kinases, which warrants further investigation.

Hence, the effect of cisplatin on Tip60 expression, SRPK1 acetylation and pre-mRNA splicing preference as proposed in the putative model (Supplementary Fig. 18) may exhibit tissue specificity, and even heterogeneity within breast tumours as suggested by HCC70 responding to cisplatin differently from the other breast cancer cell lines tested regarding SRPK1 acetylation (Fig. 2i; Supplementary Fig. 15a). Nevertheless, our study supports the argument that although Tip60 inhibitors have been developed to treat various cancer cells[54], this strategy needs to be adopted with extreme caution since Tip60 depletion could confer cisplatin resistance in both normal and cancerous mammary epithelial cells[55], and a low Tip60 level is associated with poorer survival of breast cancer patients (Supplementary Fig. 16b). In fact, some HDAC inhibitors, such as SAHA, belinostat and panobinostat, have been shown to improve the outcome of chemodrugs in preclinical and clinical tests[56]. Hence it will be of great interest to explore in future whether these inhibitors could enhance SRPK1 acetylation to reverse the pro-survival splicing in cisplatin-resistant cells.

Our data has suggested an inverse correlation between SRPK1 acetylation and phosphorylation (Fig. 3), as also reported in other proteins[57,58]. This could explain the suppressed nuclear translocation of SRPK1 in cisplatin-treated 231 and MCF7 cells (Fig. 4a–c; Supplementary Figs. 8, 9) that exhibit high levels of SRPK1 acetylation (Fig. 2i). It could be also due to the compromised stability of SRPK1 when it is acetylated (Fig. 5). Future study will be needed to address whether this is achieved through the proteasome-dependent degradation, as in the case of acetylated SRSF2[28].

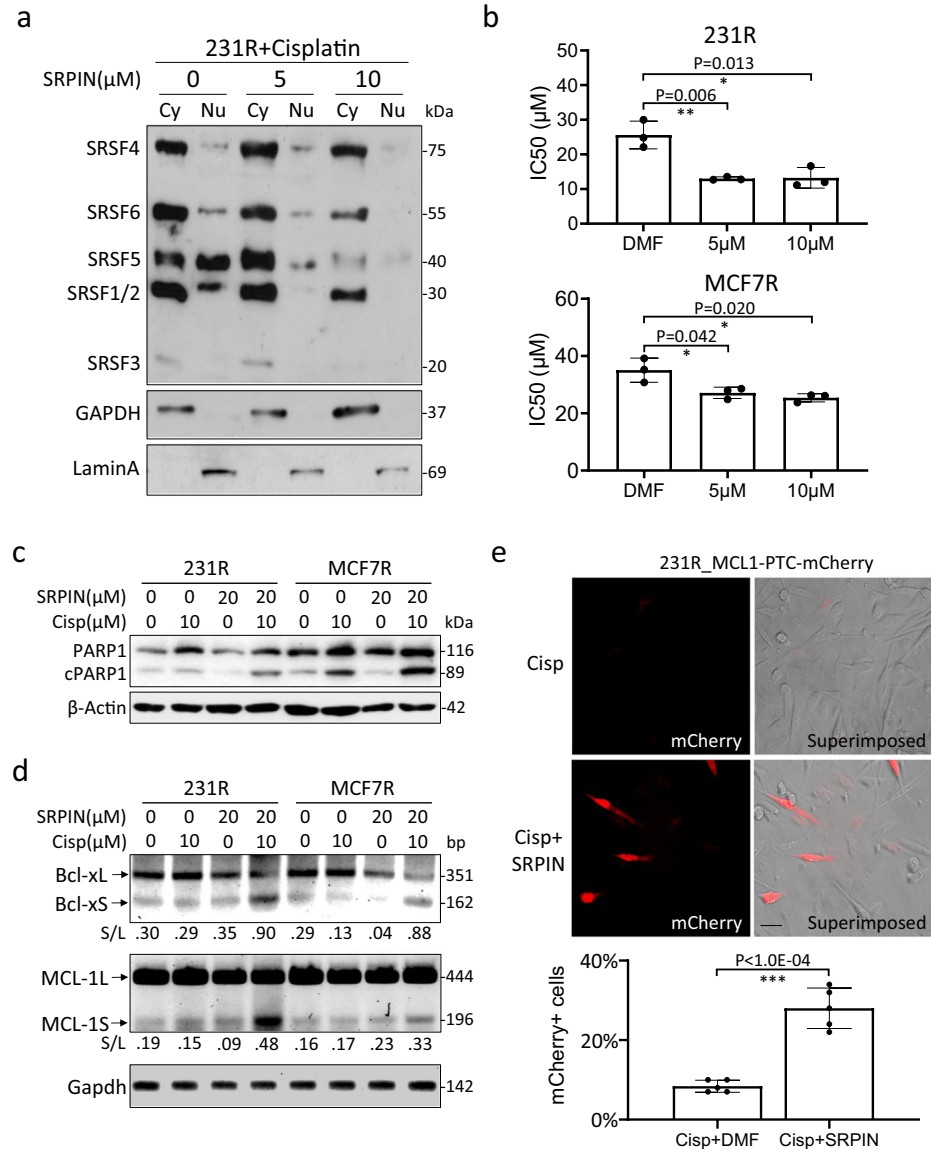

**Fig. 8 Inhibition of the SRPK1 kinase activity sensitized cells to cisplatin. a** 231R cells were co-treated with 10 μM cisplatin and indicated concentrations of SRPIN340 (SRPIN). The cells were then fractionated into the cytoplasmic (Cy) and nuclear (Nu) portions for immunoblotting with mAb104. **b** 231R and MCF7R cells were treated with indicated concentrations of SRPIN340. The IC50 of cisplatin was then determined by MTS viability assays. Bars: mean ± SD; $n = 3$; *$p < 0.05$, **$p < 0.01$ as determined by Student's $t$-test. **c, d** 231R and MCF7R were co-treated with cisplatin and SRPIN340 as indicated. The cleavage of PARP1 was assessed by immunoblotting (**c**), and the alternative splicing of *BCL2L1* and *MCL-1* by RT-PCR (**d**). The decimals below the gel strips in (**d**) denote the relative abundance of short (S) versus long (L) variants. **e** 231R cells were transfected with the splicing-sensitive reporter, MCL1-PTC-mCherry, and treated with cisplatin alone or together with SRPIN340. The mCherry signals were recorded by the fluorescence microscopy and superimposed onto the phase-contrast images. Scale bar: 20 μm. Bars: mean ± SD; $n = 5$; ***$p < 0.01$ by Student's $t$-test. The Western blots in (**a, c, d**) are representative of three experiments with similar results.

It is interesting to note that 231 cells which represent the TNBC phenotype express a higher level of SRPK1 than MCF7 cells which are hormone receptor-positive (Fig. 1c). In fact, SRPK1 is more abundantly expressed in invasive breast carcinoma[20]. However, with the development of cisplatin resistance, SRPK1 expression became higher in MCF7R than MCF7, but lower in 231R than 231 (Fig. 1c). As both hyperphosphorylation and hypophosphorylation of SRSFs will compromise pre-mRNA AS[59], an equilibrium between SRPK1 expression and SRSF phosphorylation is critical to drive the production of pro-survival splice variants to confer cisplatin resistance.

Therefore, targeting the PTMs and kinase activity of SRPK1 has the potential to re-sensitize breast cancer cells to

chemotherapy, and this strategy has the advantage of being titratable and reversible as compared with other approaches, such as genetic manipulation. Indeed, SRPIN340 was able to re-sensitize the cisplatin-resistant cells, possibly by favouring the splicing of pro-apoptotic variants (Fig. 8). The inhibitor has also been shown to reduce angiogenesis[46,60] and suppress the growth of breast and prostate cancers[16,41]. Another newly identified SRPK1/2 inhibitor, SRPIN-1, switches the splicing of VEGF towards the anti-angiogenic isoform in a neovascularization murine model[61]. In future, a genome-wide profiling of pre-mRNA AS in cisplatin-resistant cells may reveal critical splice variants that can be targeted by splice-switching oligonucleotides to suppress tumour growth as experimented previously[62–65].

In summary, our study has revealed that in at least some breast cancer cells, loss of Tip60-mediated SRPK1 acetylation may ultimately lead to anti-apoptotic AS events and predict poor outcome following cisplatin treatment (Supplementary Fig. 18). Given that cisplatin-resistant cells often exhibit cross-resistance to carboplatin and even oxaliplatin[3], future studies will be needed to explore whether by manipulating the PTMs of SRPK1, we can universally re-sensitize the resistant breast cancer cells to these compounds.

## Methods

**Antibodies**. The following antibodies were used for immunoblotting, immunocytochemistry and immunoprecipitation: anti-SRPK1 (611072, BD Biosciences), anti-Hsp90 (610418, BD Biosciences), anti-Tip60 (DR1041, Calbiochem), anti-CLK1 (ab74044, Abcam), anti-myc (#2276, Cell Signalling Tech.), anti-Flag (#14793, Cell Signaling Tech.), anti-PARP (#9542, Cell Signaling Tech.), anti-phospho-H2AX (Ser139) (#9718, Cell Signaling Tech.), anti-β-actin (sc-47778, Santa Cruz), anti-GAPDH (#5174, Cell Signaling Tech.), anti-14-3-3β (sc-59419, Santa Cruz) and anti-Lamin A (sc-6214, Santa Cruz). The antibodies were diluted 1:100 for immunofluorescence and immunoblotting. For Monoclonal antibody MAb104, hybridoma cells (ATCC® CRL-2067™) were raised in 10% FBS-containing RPMI, and the conditioned medium was collected every two days, which could be directly used for immunoblotting.

**Cell Culture, transfection and drug treatment**. MCF7 and MDA-MB-231 were cultured in High Glucose DMEM (HyClone) supplemented with 10% FBS (v/v). HCC70 and BT-549 were cultured in RPMI (HyClone) supplemented with 1 mM sodium pyruvate and 10% FBS. MDA-MB-468 was cultured in DMEM/F12 1:1 (HyClone) supplemented with 10% FBS. All the cells were maintained in 5% $CO_2$ at 37°C. To establish cisplatin-resistant cell lines, MCF7 and MDA-MB-231 cells were treated with increasing doses of cisplatin (from 1 to 10 μM) for six months; MDA-MB-468 and BT549 cells with 0.5–2.5 μM cisplatin for three months. The parental cells were treated with DMF in parallel to ensure comparability.

The transfection was conducted using Lipofectamine 3000 (Thermo Fisher) as instructed by the manual. The transfection was maintained for 72 h before the cells were collected for the downstream experiments. If significant cytotoxicity was observed, the transfection medium would be replaced after 24 h. The siRNA SMARTpools targeting human SRPK1 (M-003982-02-0010) and Tip60 (L-006301-00-0010) were purchased from Dharmacon.

Cisplatin (Sigma) and SRPIN340 (Sigma) were dissolved in DMF to make a 20 mM stock solution. The HDAC inhibitor Trichostatin A (Sigma) and the serine/threonine phosphatase inhibitor Calyculin A (Sigma) were dissolved in 100% ethanol and DMSO respectively to make a stock solution of 5 mM. For immunoprecipitation experiments, cells were treated with 20 μM (MCF-7 and MB231) /10 μM (MB468 and BT549) cisplatin or DMF for 2 days before harvest. In other experiments, 10 μM (MCF-7 and MB231)/5 μM (MB468 and BT549) cisplatin was applied for 5 days before cells were collected.

**Cell fractionation**. The cells were collected and lysed using the PARIS™ Kit (Ambion) as instructed by the manual. In brief, following centrifugation at 1000 rpm at 4 °C for 3 min, the cytoplasmic fraction (supernatant) was removed. The pellet representing the nuclear fraction was then washed twice with the cell fractionation buffer before being suspended in cell disruption buffer. A brief sonication was applied to homogenize and solubilize the pellet. The debris was removed from the nuclear lysate by centrifuging at 15,000 rpm for 5 min at 4 °C. The fractionated lysates were then analysed by Western blotting. GAPDH and Lamin A were used as the reference markers for cytoplasmic and nuclear fractions, respectively.

**Immunocytochemistry**. $2 \times 10^4$ cells were seeded on coverslips in 24-well culture plates. The treated cells were fixed with 4% paraformaldehyde, washed and blocked with 1% BSA followed by incubation with primary antibodies (1:100 in 0.1% BSA) for 90 min at room temperature. After washing three times with PBS, the cells were then incubated with secondary antibodies against mouse or rabbit that are conjugated with Alexa Fluor® 555 or Alexa Fluor 488 (Thermo Fisher) (1:200 in 0.1% BSA) for 1 hour at room temperature. The cells were counterstained with Hoechst 33342 (Sigma) (1:10,000 in PBS) and rinsed briefly with PBS. The coverslips were mounted using the ProLong Gold Antifade medium (Molecular Probes). The slides were then viewed under the CKX53 Inverted Microscope (Olympus), and images were captured using Standard CellSens (Olympus).

**Immunoprecipitation and co-immunoprecipitation**. The cells were lysed in a lysis buffer (50 mM pH 8.0 Tris-HCl, 150 mM NaCl, 1% NP-40, 0.05% sodium deoxycholate) supplemented with cOmplete™ Protease Inhibitor (Roche) for 30 min at 4 °C. The supernatant was collected after centrifugation and precleared with Protein G Plus slurry (Santa Cruz). After 10% of the precleared lysate was reserved as input, the remaining supernatant was aliquoted and incubated with 20 μl of either control IgG-coated beads (BD Biosciences) or anti-Ac-K (PTM BioLabs) or anti-phospho-Ser (Sigma) conjugated beads at 4 °C overnight. For co-immunoprecipitation, 2 μg antibodies were added to the lysate after preclearing. The incubation was performed overnight at 4 °C. The next day, 50 μl Protein G plus slurry were added to the mix and incubated for another 4 hours. Then the antibody-conjugated beads were washed sequentially with the lysis buffer, high salt buffer (50 mM pH 8.0 Tris-HCl, 500 mM NaCl, 0.1% NP-40) and low salt buffer (50 mM pH 8.0 Tris-HCl, 0.1% NP-40). The immunoprecipitates were eluted and denatured in 2×NuPAGE LDS sample buffer with 50 mM dithiothreitol (Sigma). After centrifugation, the supernatant was collected and analysed by Western blotting.

**Mass spectrometry**. The experiment was performed as described previously[24,66]. In brief, 293T cells were transfected with Myc-tagged SRPK1 and wild-type Tip60 or HAT-deficient Tip60. The cells were then lysed, and immunoprecipitation was performed with the c-Myc antibody and Protein A/G agarose beads (Santa Cruz). The precipitated proteins were subjected to trypsin digestion (protein:trypsin in a weight ratio of 1:50) overnight at 37 °C. After extraction, the digested peptides were desalted by using Aspire RP30 desalting columns (Thermo Scientific) in vacuum. Subsequently, they were analysed by the high-pressure liquid chromatography tandem mass spectroscopy (LC-MS/MS). The collected data were analysed using MASCOT (Matrix Sciences) and Protein Pilot 4.0 (ABSCIEX) for peptide and modification identifications. Carbamidomethyl (C) was set as the fixed modification; acetylation (K), oxidation (M) and deamidation (N) were set as variable modifications.

**MTS assay and determination of IC50**. $5 \times 10^3$ cells were seeded into each well in 96-well plates one day before the addition of cisplatin. Then the culture was maintained for up to 5 days before the viability assessment with CellTiter 96® AQueous One Solution Kit (Promega) as per manufacturer's instruction: 20 μl reagent was added to the medium for a 2-hour incubation. Then the absorbance at 490 nm was measured by a TECAN Infinite 200 PRO spectrophotometer. Based on the absorbance and the dose of cisplatin, the curve was plotted using the Hill equation-based regression[67]. IC50 was calculated from the regression equation where the absorbance is 50% of the maximum reading.

**Plasmid and shRNA construction**. SRPK1 was amplified by PCR using the cDNA of HeLa cells as template and cloned into pCMV-myc (N-terminal) (Clontech) or EGFP-C1 (Clontech) vector using the EcoRI and XhoI sites. Tip60 (KAT5) was cloned into pcDNA3.1(+)-3X FLAG (Thermo Fisher) using the HindIII and EcoRI sites. The mutants of SRPK1 and KAT5 were constructed using the site-directed mutagenesis, in which case the primers with mutated base pairs were designed from the opposite direction, which is 3′–5′. The PCR reactions were then subjected to the treatment of KLD enzyme mix (NEB) (containing T4 PNK, T4 DNA ligase and DpnI) for 1 hour at room temperature. DpnI was introduced as it can digest methylated DNA that is only present in plasmids but not in PCR products. All the mutants generated were validated by Sanger sequencing.

The backbone of shRNA is PLKO.1-TRC (Addgene #10878). shRNA sequences targeting human SRPK1 were designed via The RNAi Consortium (TRC) portal (http://www.broadinstitute.org/rnai/public/). Besides, the XhoI site in the shRNA loop (CTCGAG) was replaced with a PstI site (CTGCAG), making it easier to be identified with PstI digestion as the PstI site is absent in the PLKO.1 backbone. The sequences are presented in the Supplementary Table 1.

The complementary oligonucleotides were first phosphorylated by T4 PNK (NEB) in T4 DNA ligase buffer containing ATP (NEB) for 30 min at 37 °C. Subsequently, the annealing reactions for each pair of oligonucleotides were carried out at 95 °C for 5 min and gradually cooled down to room temperature using a Thermocycler (Bio-Rad), which enables the reaction to ramp down to 25 °C at 5 °C/min. At the same time, PLKO.1 TRC vector was digested with AgeI and EcoRI to create the overhangs before gel purification. Then, 1 μL annealed product was used to ligate to at least 50 ng of digested vector before transformation was performed in XL-10 Gold Competent Cells (Invitrogen).

The MCL1 minigene reporter was a gift from Prof. Pamela A. Silver, Harvard University[45].

**Quantitative and reverse-transcription PCR**. Total RNA was extracted from collected cell samples using Trizol (Invitrogen). Up to 5 μg of extracted RNA was quantified by Nanodrop One (Thermo Fisher) and incubated with Turbo DNase I (Thermo Fisher) for 30 min prior to reverse transcription using the GoScript Reverse Transcription System (Promega). Reverse transcribed cDNA was subsequently analysed on a 7900HT Fast Real-Time PCR System (Applied Biosystems) using GoTaq qPCR Master Mix (Promega). The data were analysed by the ΔΔCt approach[68] with β-actin as the internal control.

To analyse the splice variants, the reverse-transcribed cDNA was amplified with Phusion® High-Fidelity DNA Polymerase (NEB) before analysing on 1–2% agarose gels. The gels were visualized by ChemiDoc™ MP imaging system (Bio-Rad). The relative abundance of short and long variants was determined by ImageJ. The primer sequences are presented in the Supplementary Table 1.

**Western blotting**. Cells were lysed with Pierce RIPA buffer (Thermo Fisher) containing the cOmplete™ Protease Inhibitor Cocktail (Roche) and Phosphatase Inhibitor Cocktail (Thermo Fisher) with brief sonication with a Bioruptor (Diagenode). The lysate was quantified by Bradford assays (Bio-Rad) before 4×NuPAGE LDS sample buffer (Invitrogen) and 50 mM dithiothreitol (Sigma) were added. Subsequently, the protein samples were separated on NuPAGE 4–12% Bis-Tris gels (Invitrogen) and transferred to the PVDF membrane (Bio-Rad) using the XCell II blot module (Invitrogen). The blot was washed with TBS-0.1%Tween and blocked with either 5% non-fat skim milk (Bio-Rad) or BSA (HyClone) for 60 min at room temperature. The BSA was used only if phosphorylated proteins were to be detected. The blot was then incubated with primary antibodies (1:100) overnight at 4 °C. The next day, after washing three times with TBS-0.1% Tween, the blot was incubated with horseradish peroxidase-conjugated secondary antibodies (1:3000) (Invitrogen). The enhanced chemiluminescence detection solution (Thermo Fisher) was used to detect the immunoreactive bands.

To perform the Phos-tag SDS-PAGE electrophoresis to differentiate phosphorylated from unphosphorylated SRPK1, SDS-PAGE gels (6%) were supplemented with 50 μM Phos-tag AAL solution (Wako) together with 50 μM $MnCl_2$ according to the manufacturer's recommendations. Samples were lysed and prepared without EDTA as per manufacturer's instructions. The gels were run using Mini-PROTEAN Tetra Cell (Bio-Rad) at 20–30 mA per gel until the dye front completely exited the gel. The gels were then washed twice in the transfer buffer supplemented with 10 mM EDTA for 10 min. The gels were then washed with normal transfer buffer for another 10 min. The proteins were transferred to a PVDF membrane and detected using the standard Western blotting procedures as described above.

**Statistics and reproducibility**. The values are presented as mean ± SD from at least three independent experiments. The difference between treatment groups was determined by using the two-tailed unpaired Student's $t$-test, and a $P$ value < 0.05 was considered statistically significant. The precise $P$-values were also shown whenever suitable. For experiments that lack statistics, they were repeated for at least three times. The exact number of biological replicates are provided in individual figure legends.

**Reporting summary**. Further information on research design is available in the Nature Research Reporting Summary linked to this article.

## Data availability

Supplementary Data 1 contains the data presented in the bar graphs of the main figures. Supplementary Data 2 includes the potential post-translational modifications identified in SRPK1. All other data are available from the corresponding author upon reasonable request.

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

## Acknowledgements

The work was supported by the Singapore MOE Tier 1 FRC grant (T1-2014 APR-01), NMRC CBRG-NIG grant (NMRC/BNIG/2028/2015), MOE Tier 1 grant R-181-000-179-114 and NUHS Seed Fund R-181-000-192-114 awarded to Q.H. We thank Prof. Pamela A. Silver (Harvard University) for the gift of MCL-1 minigene reporter. We thank Prof. Gerald B. Call and Prof. Sudhindra R. Gadagkar (Midwestern University) for the Excel macro template for IC50 calculation.

## Author contributions

C.W. performed most of the experiments and data analysis. Z.Z. and X.F. initiated the project and identified the potential acetyltransferase for SRPK1. C.S.S., Q.C. and Z.S.L.H. provided technical support for cell culture and Western blotting. W.L. performed mass spectrometry analysis of SRPK1 acetylation. Q.H. planned and supervised the project. The manuscript was written by C.W., and edited by X.F. and Q.H.

## Competing interests

The authors declare no competing interests.
