## [Peer Review File · Communications Biology]

Reviewers' comments:

Reviewer #1 (Remarks to the Author):

Summary:

There is increasing evidence that resistance to standard cancer therapies (such as cisplatin treatment) can arise due to perturbations in the alternative splicing machinery. The present study draws attention to the splice factor kinase SRPK1, and demonstrates that SRPK1 is implicated in drug resistance. It also links a post-translational modification of SRPK1, namely its acetylation, to its activity in this context. This is an important study that consolidates the involvement of alternative splicing in drug resistance in cancer therapy, drawing attention to the splice factor kinase SRPK1 and significantly increasing our understanding of the molecular biology of SRPK1.

General comments:

1. The range and scope of experiments is commendable. However there are several results where the reproducibility of the findings is not clear, because experimental repeats are not included (see specific comments). To strengthen the key messages of the paper I would recommend incorporating, where possible, additional data obtained from experimental repeats. (This is particularly important in the alternative splicing data, the PCRs).
2. The analysis of the effect of the manipulations on alternative splicing is relatively limited to standard PCR analysis of Bcl-x and Mcl-1. Yes these are important genes implicated in apoptosis which is certainly relevant to the paper's narrative, but I would suggest that the message of the paper would be greatly enhanced with evidence of more widespread alternative splicing alterations. For example, a key cancer-associated gene whose alternative splicing is affected by SRPK1 activity is VEGF-A and it should be examined; as well as the alternative splicing of other genes that are associated with cisplatin resistance.

Specific comments:

1. P3 line 57. Although the SRPKs and CLKs are the most widely studied splice factor kinases, there are several other splice factor kinases, eg the DYRKs, PKA etc. So it is not correct to state that there are two splice factor kinase families.
2. Fig 1c, there appears to be a marginal reduction of SRPK1 following cisplatin treatment in the MCF7 and MCF7R lines, and in contrast a slight increase in treated 231 cells. Due to the marginal nature of the change here, it would be helpful to include repeats allowing quantification of the WB signal.
3. Figure 2d; the Tip60 signal in the siTip60 lane is totally absent; I find this quite unusual as generally there is some residual expression even with the most effective siRNAs. So could the authors also include repeats with a quantification of the WB signal.
4. Fig 4e; it is hard to relate the images of single cells with the quantification of nuclear K1 shown below. Could a wider field of cells be shown, and ideally benefiting from improved resolution and including a cytoplasmic marker.
5. Figure 6ab. The quantification of relative isoform levels by standard PCR is not generally very precise. Experimental repeats need to be included here in order to justify the validity of S/L ratios shown. The same point applies to the PCR data shown in Figure 8.

6. Quantification of the data shown in Fig 8e (mCherry signal) needs to be quantified.

Reviewer #3 (Remarks to the Author):

In this study, Cheng Wang et al. studied the mechanism of SRPK1 acetylation. They aimed to reveal its role in regulating cisplatin resistance in breast cancer cells. They showed some anticipated data, which might provide potential value for overcoming chemotherapy resistance. However, the quality of the paper in current stage is not good enough for publication.

1. SRPK1 expression is associated with either cisplatin sensitivity or resistance in other cancers, such as the ovarian cancer; And the acetylation of SRPK1 also has been reported previously. It greatly reduce the novelty of the work.
2. Only two different cell lines, MCF7 and 231, were utilized to obtain the conclusions. That is not enough, at least, some important results should be performed in tumor samples and animal models.
3. In figure 4 and figure 5, single cell imaging only suitable for showing details, but lack of statistical evidences.

Reviewer 1:

Summary:

There is increasing evidence that resistance to standard cancer therapies (such as cisplatin treatment) can arise due to perturbations in the alternative splicing machinery. The present study draws attention to the splice factor kinase SRPK1, and demonstrates that SRPK1 is implicated in drug resistance. It also links a post-translational modification of SRPK1, namely its acetylation, to its activity in this context. This is an important study that consolidates the involvement of alternative splicing in drug resistance in cancer therapy, drawing attention to the splice factor kinase SRPK1 and significantly increasing our understanding of the molecular biology of SRPK1.

General comments:

1. The range and scope of experiments is commendable. However, there are several results where the reproducibility of the findings is not clear, because experimental repeats are not included (see specific comments). To strengthen the key messages of the paper I would recommend incorporating, where possible, additional data obtained from experimental repeats. (This is particularly important in the alternative splicing data, the PCRs).

Thanks for the constructive suggestion. In the revised manuscript, we have provided the representative repeats for the following experiments:

- The expression of SRPK1 in the parental and cisplatin-resistant cells: **Fig. 1c** and **Sup Fig. 2a**
- Splicing of BARD1 in MDA-MB-231/MDA-MB-231R (231/231R) cells in a dose-dependent manner: **Sup Fig. 11b** and **Sup Fig. 14c**
- Effect of SRPK1 acetylation on BARD1 splicing in 231R cells: **Sup Fig. 11c, d** and **Sup Fig. 12c**
- Effect of SRPK1 acetylation on Bcl-x and MCL-1 splicing in 231R cells: **Fig. 6b** and **Sup Fig. 12c**
- Effect of SRPK1 acetylation on Bcl-x and MCL-1 splicing in MCF7R cells: **Sup Fig. 13c** and **13d**
- Effect of SRPIN340 on Bcl-x and MCL-1 splicing: **Fig. 8d** and **Sup Fig. 17**

2. The analysis of the effect of the manipulations on alternative splicing is relatively limited to standard PCR analysis of Bcl-x and Mcl-1. Yes these are important genes implicated in apoptosis which is certainly relevant to the paper's narrative, but I would suggest that the message of the paper would be greatly enhanced with evidence of more widespread alternative splicing alterations. For example, a key cancer-associated gene whose alternative splicing is affected by SRPK1 activity is VEGF-A and it should be examined; as well as the alternative splicing of other genes that are associated with cisplatin resistance.

We agreed that VEGF-A is a well-established example for the splicing regulation by SRPK1, and hence examined the splicing of VEGF-A in both 231/231R cells and MCF7/MCF7R cells. Surprisingly, the cisplatin treatment had no significant effect on VEGF-A₁₆₅ itself, and we did

not note the presence of the alternative splice variant VEGF-A_{165b} either (**Sup Fig. 14c, d**). It is possible that VEGF-A_{165b} could be expressed at an undetectable level in our settings, and other breast cancer cells can be used in future studies to identify the contribution of VEGF-A₁₆₅ splicing to cisplatin resistance.

We also examined the splicing pattern of another pro-apoptotic member of the BCL2 family, BAX, and made three observations. Firstly, the level of prototypical BAX α was higher in the parental MCF7, 231 and MDA-MB-468 cells than in the respective resistant cells (**Sup Fig. 14c, d; Sup Fig. 15f**). Secondly, the constitutive active BAX β was induced by cisplatin only in 231 cells, but not in 231R (**Sup Fig. 14c**), which could partially explain the cisplatin-induced cell death of 231. Thirdly, BAX δ , which lacks the BH3 domain for homodimerization and heterodimerization, was constitutively expressed in all the cell lines tested except 231R (**Sup Fig. 14c, d; Sup Fig. 15f**). As the exact function of BAX δ remains elusive, it will be of interest to characterize this specific BAX splice variant in future endeavours.

Specific comments:

1. P3 line 57. Although the SRPKs and CLKs are the most widely studied splice factor kinases, there are several other splice factor kinases, eg the DYRKs, PKA etc. So it is not correct to state that there are two splice factor kinase families.

Thanks for pointing out the error. We have corrected the sentence as suggested.

2. Fig 1c, there appears to be a marginal reduction of SRPK1 following cisplatin treatment in the MCF7 and MCF7R lines, and in contrast a slight increase in treated 231 cells. Due to the marginal nature of the change here, it would be helpful to include repeats allowing quantification of the WB signal.

Thanks for the suggestion. In the revised manuscript, we have included the experimental repeat of **Fig. 1c** in **Sup Fig. 2a**. The densitometry analysis showed that cisplatin decreased the expression of SRPK1 in MCF7, but increased it in 231 cells (**Sup Fig. 2b**).

3. Figure 2d; the Tip60 signal in the siTip60 lane is totally absent; I find this quite unusual as generally there is some residual expression even with the most effective siRNAs. So could the authors also include repeats with a quantification of the WB signal?

We fully understand the Reviewer's concern. As **Fig. 2c** (**Fig. 2d** in the initial submission) reported siRNA verification by RT-PCR, the absence of Tip60 band in the siTip60 lane could be due to insufficient exposure time in the Gel Doc. To further confirm the knockdown efficiency of siTip60, we have performed the quantitative PCR (**Fig. 2d**) and Western blot with densitometry (**Fig. 2e**), which showed that the siRNA can reduce the Tip60 expression by ~80% to 90%.

4. Fig 4e; it is hard to relate the images of single cells with the quantification of nuclear K1 shown below. Could a wider field of cells be shown, and ideally benefiting from improved resolution and including a cytoplasmic marker?

As suggested by the Reviewer, we have included wide-field images of live HeLa cells transfected with GFP-tagged wild-type SRPK1 and acetylation-deficient Mut7 (**Sup Fig. 9a**). To confirm the nuclear localization preferred by Mut7, we transfected MCF7 cells with the above GFP-tagged constructs, and counterstained the nuclei with Hoechst. As can be seen in **Sup Fig. 9b**, Mut7 was more prone to be present in the cell nuclei than the wild-type counterpart.

5. Figure 6ab. The quantification of relative isoform levels by standard PCR is not generally very precise. Experimental repeats need to be included here in order to justify the validity of S/L ratios shown. The same point applies to the PCR data shown in Figure 8.

As mentioned in the response to the General Comment 1 above, we have included the following experimental repeats:

- Splicing of BARD1 in 231/231R cells in a dose-dependent manner: **Sup Fig. 11b** and **Sup Fig. 14c**
- Effect of SRPK1 acetylation on BARD1 splicing in 231R cells: **Sup Fig. 11c, d** and **Sup Fig. 12c**
- Effect of SRPK1 acetylation on Bcl-x and MCL-1 splicing in 231R cells: **Fig. 6b** and **Sup Fig. 12c**
- Effect of SRPK1 acetylation on Bcl-x and MCL-1 splicing in MCF7R cells: **Sup Fig. 13c** and **13d**
- Effect of SRPIN340 on Bcl-x and MCL-1 splicing: **Fig. 8d** and **Sup Fig. 17**

6. Quantification of the data shown in Fig 8e (mCherry signal) needs to be quantified.

As requested by the Reviewer, we have quantified the mCherry-positive cells in **Fig. 8e**.

We would like to thank Reviewer 1 for the critical reading of the manuscript and constructive suggestions.

Reviewer 2:

Disturbed alternative splicing is a common feature in human tumors. SRPK1 is a key kinase that phosphorylates the serine/arginine (SR)-rich domains of splicing factors, thus regulating their subcellular localization and interactions with RNA and protein partners. SRPK1 has been reported to be dysregulated in multiple cancers, yet its function is complicated and contradictory, since it acts as both tumor suppressor and promoter. Furthermore, SRPK1 has been associated both with cisplatin sensitivity and resistance. In this manuscript the authors provide an explanation on the controversial relation of SRPK1 to cisplatin responsiveness, based on an inverse correlation between two post-translational modifications of the kinase. More specifically, the authors found that in breast cancer cells (MCF7 and 231) SRPK1 becomes acetylated in a Tip60-dependent manner upon treatment with cisplatin, while it remains hypophosphorylated. On the contrary, in the corresponding resistant cells, cisplatin reduces acetylation but increases phosphorylation and kinase activity of SRPK1. While acetylation/hypophosphorylation of SRPK1 favors the splicing of pro-apoptotic variants, loss of Tip60-mediated acetylation and increased phosphorylation lead to anti-apoptotic splicing

events. Cisplatin-resistant cells could be re-sensitized by enhancing SRPK1 acetylation or inhibiting its kinase activity. Given the importance of cisplatin and other platinum-based compounds to treat breast cancer, the elucidation of a molecular mechanism that modulates drug responsiveness is of great interest to a large audience. However, the manuscript presents some flaws that are summarized below.

1. Tip60 has classically been thought of as a nuclear protein (see for example <https://www.uniprot.org/uniprot/Q92993> (KAT5/Tip60) or image provided by abcam for the anti-KAT5/Tip60 antibody). Yet, the main assumption underlying the authors' model (Supplementary Figure 7) is that Tip60 localizes to the cytoplasm. There is no evidence provided throughout the manuscript that supports the cytoplasmic localization of Tip60 in MCF7, MCF7R, 231 and 231R cells. Supplementary figure 3 shows a western blotting analysis in which Tip60 is rather equally distributed between the nucleus and the cytoplasm in HeLa cells. However, the nuclear localization of Tip60 in HeLa cells is well established (see for example Ashraf et al. (2017) Journal of Experimental and Clinical Cancer Research; Fig. 6 Expression and localization of Tip60 mutants in HeLa cells). The authors, in order to substantiate their model, need to provide evidence on the cytoplasmic localization of Tip60 in MCF7 and 231 cells. By the way, why did the authors choose HeLa cells to check the subcellular localization of Tip60, while MCF7 and 231 cells have been used throughout the manuscript?

We agree with the Reviewer that Tip60 has been classically regarded as a nuclear protein. For **Sup Fig. 3** (now **Sup Fig. 9c** in the revised manuscript), the aim was to examine the effect of Tip60-induced acetylation on SRPK1 subcellular localization. HeLa cells were chosen mainly for the amenability to transfection. In fact, we were also surprised by the cytoplasmic localization of Tip60 suggested by the Western blot. A possible explanation is that overexpression may distort the localization of the exogenous Flag-tagged Tip60.

As such, in the revised manuscript, we checked the subcellular distribution of endogenous Tip60 in both MCF7 and MDA-MB-231 (231) cells. We found that although the HAT was dominantly localized in the nuclei, a substantial amount of the protein was present in the cytoplasm (**Sup Fig. 5b**). Interestingly, we also noted some reports describing the presence of Tip60 outside the cell nucleus:

- Hass & Yankner, 2005, J Biol Chem. 2005, 280(44):36895-904. [1]
- Bassi et al., Cell Death Differ. 2016, 23(7):1198-208. [2]
- McGuire et al., Sci Rep. 2019, 9(1):3819. In this report, cytoplasmic localization of Tip60 was found prevalent in a breast cancer tissue microarray. [3]

Hence, it is conceivable that the subcellular distribution of Tip60 may be quite dynamic and exhibit cell, tissue and even disease specificity.

2. The authors mention that SRPK1 is subject to autophosphorylation. However, it was shown (ref. 24) that this autophosphorylation event practically occurs only in the presence of

activated Akt, which in turn is activated by extracellular signals, such as EGF signaling. Do the authors imply that cisplatin treatment of resistant cells mimics EGF signaling? In such a case, they should present evidence on Akt activation. Furthermore, Akt-induced autophosphorylation of SRPK1 does not affect kinase activity but induces nuclear translocation of SRPK1 (ref.24). Contrary to these data, it is clearly shown in Figure 3 that reduction of SRPK1 acetylation led to elevated phosphorylation and kinase activity towards SR splicing factors, while only a slight nuclear translocation of SRPK1 was observed (see also comment below). SRPK1 is considered as a constitutively active kinase. There is only one report in the literature (Mylonis and Giannakouros *Biochem. Biophys. Res. Commun.* 2003), showing that CK2 phosphorylates and moderately activates SRPK1. Do the authors imply that cisplatin somehow promotes CK2-mediated phosphorylation of SRPK1 in resistant cells? The authors should further clarify the phosphorylation event observed by cisplatin treatment/acetylation deficiency.

Thanks for the very helpful suggestion. In the revised manuscript, we examined the activation of AKT and as shown in **Sup Fig. 7**, the phosphorylation level of AKT-Ser473 was higher in the cisplatin-resistant 231 cells than in the parental cells. However, the phosphorylation of mTOR-Ser2448 was not consequently elevated in such 231R cells, suggesting that AKT signalling may indeed branch to SRPK1 as reported (Zhou et al., 2012) [4].

As for SRPK1 subcellular distribution, we have now shown that the bulk of SRPK1 was localized in the nuclei of MCF7 cells while a substantial amount of the kinase was present in the nuclei of MCF7R (**Fig. 4a**). A significant accumulation of SRPK1 was also observed in the nuclei of 231R cells as compared to those of 231 by immunocytochemistry and immunoblotting (**Sup Fig. 8a, 8b**). Furthermore, cisplatin was found to reduce the nuclear abundance of SRPK1 in 231 cells (**Sup Fig. 8b**), in which SRPK1 acetylation was increased by the treatment (**Fig. 3f**).

However, in 231R cells, cisplatin did not further increase the nuclear abundance of SRPK1 as compared to the DMF solvent (**Sup Fig. 8b**), which seems to contradict with a high level of SRPK1 phosphorylation (**Fig. 3f**) as pointed out by the Reviewer. This could be due to several factors. Firstly, the resistant line was established by exposing to cisplatin for six months, and then maintained in the presence of cisplatin for the entire duration of the project. Hence, the cells may have already evolved to have a higher abundance of SRPK1 in the nuclear compartment as compared to the parental cells. This is indeed observed for DMF-treated MCF7R (**Fig. 4a**) and 231R (**Sup Fig. 8a**). Yet when we compared the cisplatin-treated parental and resistant cells, we noted that the latter had a lower level of SRPK1 acetylation (**Fig. 2i; Fig. 3f**), a higher level of phosphorylation (**Fig. 3c, 3f**) and a more prominent nuclear accumulation of the kinase (**Fig. 4a; Sup Fig. 4a, 4b**).

Secondly, an excessive amount of nuclear SRPK1 could de-regulate the splicing process and cause cell death in several ways: 1) it may prevent de-phosphorylation of SRSFs, which is essential for their nuclear export (Cao et al., 1997; Prasada et al., 1999) [5, 6]; 2) it could result in insufficient amount of the kinase in the cytoplasm to re-phosphorylate the recycled SRSFs

(Gilbert & Guthrie, 2004; Ding et al., 2006) [7, 8]; 3) excessive SRPK1 may interfere with the stripping of CLK1 from the fully phosphorylated SRSFs in the nucleus (Aubol et al., 2016) [9]. Hence, the resistant cells may have reached a balance of having more SRPK1 in the nucleus than the parental cells without jeopardizing the splicing machinery and cell survival.

3. According to the authors the inverse correlation between SRPK1 acetylation and phosphorylation (Fig. 3) could explain the suppressed nuclear translocation of SRPK1 in cisplatin-treated 231 and MCF7 cells (Fig. 4a-d; Supplementary Fig. 3a). At first, the authors should provide an immunofluorescence image of control 231 and MCF7 cells to substantiate their conclusion that the nuclear translocation of SRPK1 was suppressed after cisplatin treatment. In cisplatin-resistant cells, acetylation is significantly reduced, whereas there is an increase of the phosphorylation levels.

However, in the author's model presented in Supplementary Figure 7 SRPK1 is shown again to have a cytoplasmic localization. Furthermore, while the authors mention that SRPK1 was more nucleus localized in cisplatin-treated resistant cells, their conclusion is barely supported by the Western blotting analysis in Fig. 4B and the immunofluorescence image of MCF7R cells (Supplementary figure 3, again staining of control MCFR cells is missing). Figure 4a is more convincing but only a couple of cells with significant nuclear staining are shown without providing any quantitative data on a substantial number of cells. In all localization studies, cells were treated with cisplatin (the concentration is not mentioned in the text) for 2 days. It is not clear from Figure 1a the effect of 2 days-cisplatin treatment on the cells. Would the results be the same if they treated cells with 10-20 μ M cisplatin for 5 days?

We are very sorry that the immunolabelling of SRPK1 in mock-treated cells was left out in the initial submission. As requested, in the revised manuscript, we have furnished the images of DMF-treated MCF7/MCF7R (Fig. 4a) and 231/231R (Sup Fig. 8a). In both cases, more SRPK1 was present in the nuclei of resistant cells. On the other hand, in the parental cells, immunofluorescence did not reveal a dramatic effect of cisplatin on reducing the nuclear accumulation of SRPK1. This could be due to an already low nuclear abundance of the kinase in parental cells, which makes it difficult to see further reduction by immunofluorescence. Nevertheless, immunoblotting showed that cisplatin could significantly reduce SRPK1 accumulation in the nuclei of 231 cells (Sup Fig. 8b).

As for the model (now Sup Fig. 18), we would like to thank the Reviewer for pointing out the confusion, and have made some modifications to show that the nuclear abundance of SRPK1 is higher in the cisplatin-resistant cells than in the parental cells.

We apologize for the inconsistency between the figure legend and Materials & Methods section in the initial submission. In all the experiments except immunoprecipitation, cisplatin was given for 5 days at 10 μ M. We have corrected the figure legends in the revised manuscript.

4. The time selected for cisplatin-treatment in the various assays of the manuscript creates confusion. The authors mention in the materials and methods section that for

immunoprecipitation experiments cells were treated with 20 μ M cisplatin for 2 days, while in all other experiments 10 μ M cisplatin was applied for 5 days. However, cisplatin (unknown concentration) was applied for 2 days in the localization assays (both immunofluorescence and immunoblotting, Figure 4, Supplementary Figure 3), 10 μ M cisplatin was applied for 2 days in the immunoblotting assays presented in Figure 2g and the time of treatment is not mentioned in the splicing and IC50 assays (Figure 6a, Supplementary Figures 4 and 5, Figure 7a, Figure 8b). The authors should treat the cells with same concentration of cisplatin and for the same time in all assays. Furthermore, Figure 1 should cover all days of cisplatin treatment (day 1 - day 5) at a given concentration of cisplatin.

Again we would like to apologize for the confusion caused by the inconsistency between the Materials & Methods section and figure legends. As stated in the Materials & Methods, for immunoprecipitation, the cells were treated with 20 μ M cisplatin for 2 days, while for the other experiments, including immunofluorescence and immunoblotting, cisplatin was given for 5 days at 10 μ M. As suggested, we have included the time-course data on the protein level of SRPK1 in MCF7/MCF7R and 231/231R cells which were treated with 10 μ M cisplatin for 1 day, 3 days and 5 days (**Sup Fig. 3**).

We would like to thank Reviewer 2 for raising very insightful questions and making helpful suggestions.

Reviewer 3:

In this study, Cheng Wang et al. studied the mechanism of SRPK1 acetylation. They aimed to reveal its role in regulating cisplatin resistance in breast cancer cells. They showed some anticipated data, which might provide potential value for overcoming chemotherapy resistance. However, the quality of the paper in current stage is not good enough for publication.

1. SRPK1 expression is associated with either cisplatin sensitivity or resistance in other cancers, such as the ovarian cancer; and the acetylation of SRPK1 also has been reported previously. It greatly reduces the novelty of the work.

We agree with the Reviewer that aberrant SRPK1 expression has been reported to cause either chemotherapy sensitivity or resistance in various cancers, such as ovarian, lung, prostate, breast and colon. In fact, we were amazed by this lack of consistency regarding the role of SRPK1 expression in cell fate determination, and hence decided to investigate whether some other aspect, such as post-translational modification (PTM), of the kinase is an important factor in this regard.

It is well established that SRPK1 itself can be phosphorylated. Interestingly, an acetylome study using acute myeloid leukaemia cells found that many proteins involved in RNA splicing and processing can also be acetylated (Choudhary et al., 2009) [10]. In particular, two lysine

residues in SRPK1, K585 and K588, were shown to be acetylated in that study. However, whether SRPK1 acetylation also occurs in other cellular contexts, such as breast cancer cells, and more important, whether acetylation affects the biological function of the kinase remain unknown.

Indeed, our experiments showed that SRPK1 could be acetylated in a Tip60-dependent manner in breast cancer cells. Moreover, we noted that this PTM of SRPK1 affected the alternative splicing of some apoptosis-related factors and potentially modulated cell responsiveness to cisplatin. Hence, our data suggest that SRPK1 acetylation may be targeted to overcome cisplatin-resistance in breast cancer cells. According to literature search, our study is the first one to characterize the biological relevance of SRPK1 acetylation.

2. Only two different cell lines, MCF7 and 231, were utilized to obtain the conclusions. That is not enough, at least, some important results should be performed in tumor samples and animal models.

We agree that the initial manuscript can be greatly strengthened by performing the key experiments in tumor samples and animal models. However, due to the funding situation and the timeline to obtain the relevant ethical approval, we have instead utilized some additional breast cancer cell lines, HCC70, BT549 and MDA-MB-468 (468), all of which represent the triple negative subtype.

From the experiments, we found that cisplatin induced upregulation of Tip60 and SRPK1 acetylation in BT549 and 468 cells (**Sup Fig. 15a-e**). Moreover, cisplatin switched the splicing towards the pro-apoptotic Bcl-xS and MCL-1S in the parental 468 cells, but not in the corresponding cisplatin-resistant line (**Sup Fig. 15f**). However, a consistent cisplatin-induced upregulation or downregulation of SRPK1 expression was not recorded in these cells (**Sup Fig. 15c, d**), again suggesting that the protein level of the kinase may not be the major determinant of cell responsiveness to cisplatin. Hence, we propose that cisplatin-induced SRPK1 acetylation could be a common event in at least some subset of breast cancer cells, and it will be of great significance, as suggested by the Reviewer, to further validate this notion in tumor samples and animal models.

3. In figure 4 and figure 5, single cell imaging only suitable for showing details, but lack of statistical evidences.

For **Fig. 4a**, we have provided the wide field images to show multiple cells which were presented as **Sup Fig. 8a** in the revised manuscript.

For **Fig. 4d**, we have supplemented it with the wide field images which were presented as **Sup Fig. 9a, 9b** in the revised manuscript.

For **Fig. 5c**, we have supplemented it with the wide field images which were presented as **Sup Fig. 10** in the revised manuscript.

We would like to thank Reviewer 3 for reading the manuscript critically and making very insightful suggestions for our future explorations.

Reference list:

1. Hass, M.R. and B.A. Yankner, *A {gamma}-secretase-independent mechanism of signal transduction by the amyloid precursor protein*. J Biol Chem, 2005. **280**(44): p. 36895-904.
2. Bassi, C., et al., *The acetyltransferase Tip60 contributes to mammary tumorigenesis by modulating DNA repair*. Cell Death Differ, 2016. **23**(7): p. 1198-208.
3. McGuire, A., et al., *Quantifying Tip60 (Kat5) stratifies breast cancer*. Sci Rep, 2019. **9**(1): p. 3819.
4. Zhou, Z., et al., *The Akt-SRPK-SR axis constitutes a major pathway in transducing EGF signaling to regulate alternative splicing in the nucleus*. Mol Cell, 2012. **47**(3): p. 422-33.
5. Cao, W., S.F. Jamison, and M.A. Garcia-Blanco, *Both phosphorylation and dephosphorylation of ASF/SF2 are required for pre-mRNA splicing in vitro*. RNA, 1997. **3**(12): p. 1456-67.
6. Prasad, J., et al., *The protein kinase Clk/Sty directly modulates SR protein activity: both hyper- and hypophosphorylation inhibit splicing*. Mol Cell Biol, 1999. **19**(10): p. 6991-7000.
7. Gilbert, W. and C. Guthrie, *The Glc7p nuclear phosphatase promotes mRNA export by facilitating association of Mex67p with mRNA*. Mol Cell, 2004. **13**(2): p. 201-12.
8. Ding, J.H., et al., *Regulated cellular partitioning of SR protein-specific kinases in mammalian cells*. Mol Biol Cell, 2006. **17**(2): p. 876-85.
9. Aubol, B.E., et al., *Release of SR Proteins from CLK1 by SRPK1: A Symbiotic Kinase System for Phosphorylation Control of Pre-mRNA Splicing*. Mol Cell, 2016. **63**(2): p. 218-228.
10. Choudhary, C., et al., *Lysine acetylation targets protein complexes and co-regulates major cellular functions*. Science, 2009. **325**(5942): p. 834-40.

REVIEWERS' COMMENTS:

Reviewer #1 (Remarks to the Author):

I would like to thank the authors for comprehensively addressing my points.

Reviewer #2 (Remarks to the Author):

The authors have made substantial improvements to the manuscript by inclusion of additional data. However, I still have some concerns based on discrepancies with previously published research that need discussion and if possible clarification.

The authors noted a cisplatin-induced increase of Tip60 in MCF7, 231, BT549 and 468 cell lines. Edmond et al. (EMBO J. 2011, the first manuscript presenting evidence on SRPKs acetylation by Tip60) reported that upon cisplatin treatment, Tip60 protein level strongly decreased in lung carcinoma and osteosarcoma cell lines. A similar strong decrease of Tip60 levels was also observed by the authors in HCC70 cells (suppl. Fig. 15).

According to Edmond et al. the decrease of Tip60 levels, following cisplatin treatment led to the nuclear accumulation of both SRPK1 and SRPK2 kinases (see Fig. 7, model). According to the authors cisplatin was found to reduce the nuclear abundance of SRPK1 in 231 cells in which SRPK1 acetylation was increased by the treatment (Fig. 3f and immunoblotting Suppl. Fig. 8b).

My first concern is whether the model presented by the authors concerning the effect of cisplatin to cancer cells is applicable only in certain cancer lines, whereas in others treatment with cisplatin results in different responses.

Furthermore, based on the observation that SRPK1 acetylation was increased by cisplatin the authors report that in 231 cells when Tip60 was knocked down, the anti-apoptotic MCL-1L was dramatically increased (Supplementary Fig. 13b, lanes 3 and 4). Thus the authors come to the conclusion that SRPK1 acetylation favours pro-apoptotic splicing, whereas phosphorylation favours anti-apoptotic signaling. While this notion makes perfect sense for the resistant cells in which SRPK1 is phosphorylated and non-acetylated, it is hard to understand for the control cells (without cisplatin treatment) in which SRPK1 is acetylated and non-phosphorylated. Does this mean that in control MCF7 or 231 cells the splicing machinery favours the production of pro-apoptotic molecules?

Reviewer #3 (Remarks to the Author):

I recommend the revised version for publication.

In this round of revision, the following changes have been made to the main text:

1. We discussed the concern raised by Reviewer 2 of whether Tip60 expression could respond to cisplatin and contribute to resistance in a disease specific manner, and emphasized the cell specificity of our putative model (Supplementary Fig. 18).
2. As suggested by Reviewer 2, we moderated the conclusion derived from Supplementary Fig. 13a regarding the impact of SRPK1 acetylation on splicing preference.